# 3D reconstruction of skin and spatial mapping of immune cell density, vascular distance and effects of sun exposure and aging

Soumya Ghose[1,4], Yingnan Ju[2,4], Elizabeth McDonough[1], Jonhan Ho[3], Arivarasan Karunamurthy[3], Chrystal Chadwick[1], Sanghee Cho[1], Rachel Rose[1], Alex Corwin[1], Christine Surrette[1], Jessica Martinez[1], Eric Williams[1], Anup Sood[1], Yousef Al-Kofahi[1], Louis D. Falo Jr[3], Katy Börner[2✉] & Fiona Ginty[1✉]

Mapping the human body at single cell resolution in three dimensions (3D) is important for understanding cellular interactions in context of tissue and organ organization. 2D spatial cell analysis in a single tissue section may be limited by cell numbers and histology. Here we show a workflow for 3D reconstruction of multiplexed sequential tissue sections: MATRICS-A (Multiplexed Image Three-D Reconstruction and Integrated Cell Spatial - Analysis). We demonstrate MATRICS-A in 26 serial sections of fixed skin (stained with 18 biomarkers) from 12 donors aged between 32–72 years. Comparing the 3D reconstructed cellular data with the 2D data, we show significantly shorter distances between immune cells and vascular endothelial cells (56 μm in 3D $vs$ 108 μm in 2D). We also show 10–70% more T cells (total) within 30 μm of a neighboring T helper cell in 3D $vs$ 2D. Distances of p53, DDB2 and Ki67 positive cells to the skin surface were consistent across all ages/sun exposure and largely localized to the lower stratum basale layer of the epidermis. MATRICS-A provides a framework for analysis of 3D spatial cell relationships in healthy and aging organs and could be further extended to diseased organs.

[1] GE Research Center, 1 Research Circle, Niskayuna, NY 12309, USA. [2] Indiana University, 107 South Indiana Ave, Bloomington, IN 47405, USA. [3] University of Pittsburgh School of Medicine, 3550 Terrace St, Pittsburgh, PA 15213, USA. [4] These authors contributed equally: Soumya Ghose, Yingnan Ju.
✉email: katy@indiana.edu; fiona.ginty@ge.com

The National Institutes of Health's (NIH) Human Biomolecular Atlas Program (HuBMAP) aims to create a comprehensive high-resolution atlas of all cells in the healthy human body using data from multiple laboratories across the US and Europe[1]. Integrating and harmonizing the data derived from these samples and "mapping" them into a common three-dimensional (3D) space is a major challenge. HuBMAP, in close collaboration with 17 other international consortia and projects, is systematically constructing a Human Reference Atlas (HRA)[2]. At the core of this Atlas is a common coordinate framework (CCF) that supports spatially and semantically explicit human tissue registration and exploration. The completed Atlas will support the design of a "digital twin" for healthy men and women that can be parameterized in support of precision health and medicine. The CCF has two key components: (1) Anatomical Structures, Cell Types, and Biomarkers (ASCT + B) tables for each organ, which utilize existing ontologies (e.g., Uberon multi-species anatomy ontology, Foundational Model of Anatomy Ontology [FMA], Cell Ontology [CL], or HUGO Gene Nomenclature [HGNC]) and (2) a 3D reference object library that spatially defines anatomical structures and cell types for each organ and characterizes their 3D spatial relationships within the human body. Using the Atlas, tissue data can be registered spatially and semantically (using standardized and unified names for anatomy, cell types, and biomarkers). This paper focuses on the generation of 3D skin data and automated computation of 3D spatial maps. To the best of our knowledge, this is the first-time immune cell cluster density in 3D, distance distributions for immune cells to the nearest endothelial cell in 3D, and distance of damaged or proliferating cells to the skin surface has been presented.

Skin is the largest human organ. It is composed of at least 36 different cell types (documented in version 1.2 of ASCT + B[3]) and a vast microenvironment of over 16 anatomical structures including glandular structures, hair follicles, vasculature, and immune system components. At least 70 protein biomarkers are commonly used to characterize major skin cell types and anatomical structures[3]. While several single-cell atlas studies of human skin have been conducted in recent years[4,5], these have focused on single-cell (sc) RNAseq analysis and not on 2D in situ or 3D spatial analysis of cell types and proteins. An in-depth proteomics analysis of healthy skin identified 10,701 proteins and used advanced dissection and flow cytometry to map them to the location within the skin layers and cellular origin[6]. While much work has been done on characterizing pre-cancerous and cancerous skin, there is less understanding of cellular changes in otherwise healthy individuals across the lifecycle, including the effects of UV[7,8]. A recent multimodal analysis (combining scRNAseq, spatial transcriptomic, and multiplexed ion beam imaging)[9] of cutaneous squamous cell carcinoma (cSCC) and matched normal skin showed largely overlapping keratinocyte populations, with just one unique tumor-specific keratinocyte subpopulation in sSCC. This highlights the value of developing normal/healthy reference organ datasets to understand the cellular transition to disease.

In recent years, there have been a growing number of investigations into the cellular biology of healthy and diseased organs in 3D[10–17]. Specifically in relation to skin, Wang et al.[18] used confocal microscopy to demonstrate blood vessel and lymphatic networks in the human dermis using immunostaining for CD31 (endothelial cells), podoplanin, and LYVE1 (lymphatic cells). Light-sheet microscopy has been used to image skin structure in 3D[10], however, the use of low numerical aperture objectives and low magnification required to achieve a wide field of view results in low spatial resolution at a cellular level. Reconstruction in 3D of serially sectioned H&E-stained skin/scalp samples using the CODA method has demonstrated variations in dermis structure

and other anatomical features, including hair follicles and vasculature[19]. In oncology applications, 3D reconstruction of colon cancer[12] combined serial H&E sections (n = 22) and serial multiplexed images (n = 25) and demonstrated features in 3D that were not evident in 2D, including the interconnectivity of histological structures in the tumor microenvironment, such as tumor buds, as well as cellular and morphology transitions and gradients. Imaging mass cytometry of serially cut tissue sections (n = 152) of breast cancer reconstructed in 3D[13] demonstrated cellular and microenvironment heterogeneity that was not measurable in 2D, including more cell–cell interactions and colocalization (e.g. pS6+ cells and SMA+ cells) and clusters of invasive cells.

In general, reconstruction of 3D volumes from 2D serial sections is a complex procedure and can suffer from the "banana effect" (where curved structures are incorrectly straightened during image registration) in the absence of external reference structures[20,21]. Further, the 3D reconstruction process tends to be computationally slow. To address these challenges, we have developed an automated, reproducible workflow (Multiplexed Image Three-D Reconstruction and Integrated Cell Spatial - Analysis - MATRICS-A) for 3D reconstruction of highly multiplexed tissue sections. Compared to previous 3D reconstruction methods[14–17], our approach is calibrated using micro CT images of the formalin-fixed block, thus improving 3D reconstruction accuracy (and reducing the "banana effect"). We demonstrate the utility of MATRICS-A in multiplexed serial sections of skin collected from younger and older donors and different anatomical regions. We provide interactive visualization tools for 3D cellular data in the skin epidermis and dermis, including immune cell cluster density, spatial distances between immune cells and nearest endothelial cells, and localization of ultraviolet (UV) radiation-damaged cells (e.g., p53 mutations), DNA repair (DDB2), proliferation (Ki67) markers and their distance to the skin surface. Comparing the 3D reconstructed data with the 2D data from each section, we show significantly shorter distances between immune cells and vascular endothelial cells (56 μm in 3D vs 108 μm in 2D). Most T helper cells were found within 25 μm of the nearest endothelial cell, hence it is possible that spatial relationships with vasculature are not fully accounted for in 2D. We also show 10–70% more T cells (total) within 30 μm of a neighboring T helper cell in 3D vs 2D. This is an important consideration in samples with low immune cell density where spatial relationships and cell–cell interaction analysis in 2D may be challenging to accurately quantify. Distances of p53, DDB2, and Ki67 positive cells to the skin surface were consistent across all ages/sun exposure and largely localized to the lower stratum basale layer of the epidermis. This agrees with previous studies that show that Ki67 and/or p53 positive cells do not increase until actinic keratosis (damage to keratinocytes in the lower third of the epidermal layer) or a pre-cancerous/cancerous (full epidermis thickness) squamous in situ state is reached[22]. MATRICS-A software provides an open access framework for 3D reconstruction of multiplexed tissue and helps increase understanding of cell–cell relationships including immune cell interactions and vascular distances in the skin, which can be extended to other organs and disease states.

## Results

**Building a skin 3D reference organ**. The end-to-end workflow for this study is shown in Fig. 1. To generate the samples needed for reconstruction we leveraged archived formalin-fixed, healthy skin biopsies from 12 donors (age range 32–72 years) (Supplementary Table 1). The samples were trimmed (~2 to 7 mm size range), preserving the epidermis and dermis structure, and

# Construction of a 3-D Cellular Map of Healthy Skin

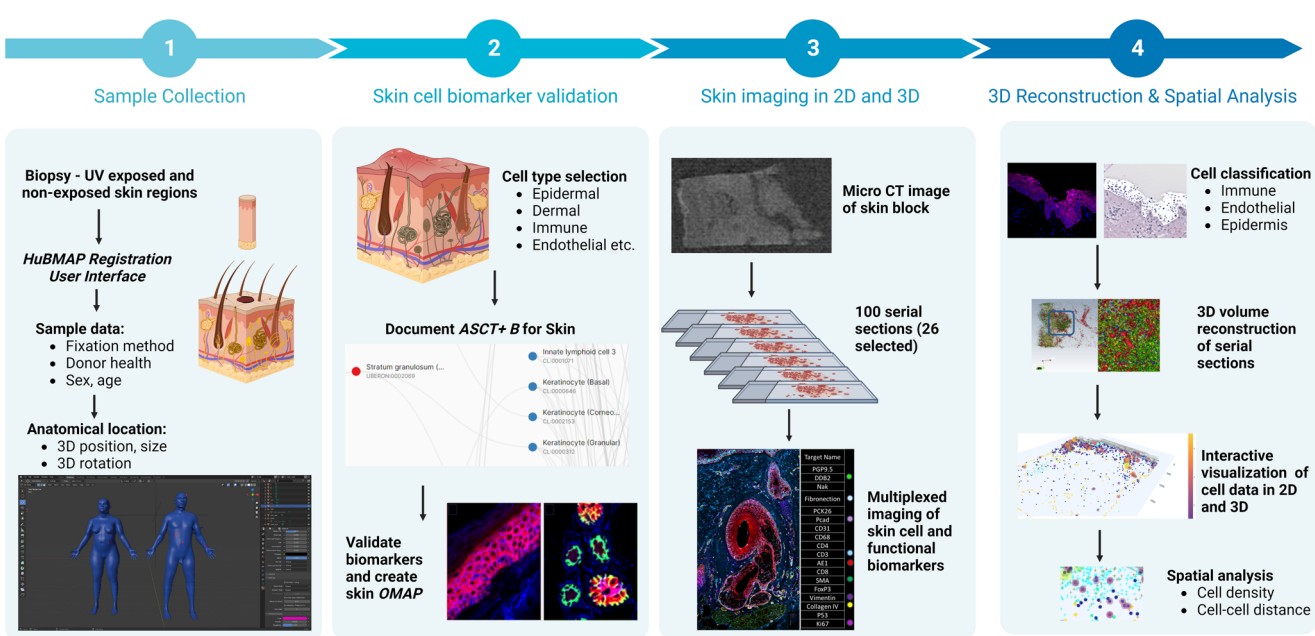

**Fig. 1 End-to-end workflow for generation of a 3D skin map of cell types and spatial distance analysis visualization tools.** (1) Healthy skin biopsies were embedded into a single formalin-fixed and paraffin-embedded (FFPE) tissue block. The human male and female skin 3D reference organ was used to spatially register and semantically annotate the biopsies via the HuBMAP Registration User Interface; (2) Skin biomarkers were identified using the skin ASCT + B tables and corresponding antibodies were validated; (3) The tissue block underwent micro CT imaging and was then sectioned into 26 serial sections for highly multiplexed immunofluorescence imaging using 18 protein and cell type markers; (4) Cell classification was conducted for each section using a hybrid supervised (deep learning-based) and unsupervised (probability-based) GMM workflow; 2D serial sections and segmented cells then underwent 3D reconstruction. The 3D spatial location of cells was used to compute immune cell cluster density, immune cell distributions from endothelial cells, and distribution and distances of p53, DDB2, and Ki67 positive cells from the skin surface. Created using BioRender.com.

embedded in a single paraffin block (layout shown in Supplementary Fig. 1). We used a human male and female skin 3D reference organ to spatially register and semantically annotate the biopsies via the HuBMAP Registration User Interface (RUI) (Supplementary Fig. 2)[23]. These reference organs were derived from the National Library of Medicine (NLM) Visible Human project[24] data and added to the HuBMAP RUI, making it possible to formally register all tissue samples into the evolving Human Reference Atlas. The resulting metadata for each tissue block documents the size, spatial location, and rotation in 3D. All registered tissue blocks used in this study can be interactively explored in the HuBMAP portal's CCF Exploration User Interface (CCF-EUI)[25].

**Construction of a skin anatomical structure and cell type + biomarker (ASCT + B) table.** In accordance with HuBMAP mission to document all cell types in the organs under investigation, an Anatomical Structure, Cell Type and Biomarkers (ASCT + B) table was constructed for the skin. The table was authored in consultation with organ experts, and it provides a reference framework for all major cell types, their anatomical location, and the protein biomarkers used to characterize these cell types. The version 1.2 skin ASCT + B master table[3] captures three critical elements: (1) the *part_of* relationships between 15 anatomical structures that are linked to their respective Uberon IDs; (2) the 36 skin cell types (linked to CL ontology) that are *located_in* one or more of these anatomical structures; and (3) 70 protein biomarkers (linked to HGNC ontology) that are commonly used to *characterize* the 36 cell types. A subset of the skin ASCT + B report is shown in Supplementary Fig. 3. The complete table with

all 36 cell types and 70 biomarkers is also accessible on our Companion Website[26] and can be interactively explored via the HuBMAP ASCT + B Reporter tool[27]. Here we focus on a subset of the skin ASCT + B table comprising 14 cell types and/or anatomical structures spanning the epidermis (stratum granulosum, stratum spinosum, and stratum basale) keratinocytes and dermis (glandular structures, fibroblasts, macrophages, T helper cells, T killer cells, T regs, nerve fibers, and endothelial cells), as well as markers of DNA damage (p53), DNA repair (DDB2), and cell proliferation (Ki67) (summarized in Supplementary Table 2). The rationale for choosing these biomarkers was to quantify (1) immune cell density in 3D vs 2D; (2) demonstrate 3D spatial relationships between immune cells and nearest endothelial cells; (3) measure the spatial cellular effects of aging and sun exposure on epidermis cells. Antibody information for each target protein is shown in Supplementary Table 3. Commercially sourced antibodies were validated and conjugated using a standardized protocol[28] (see Methods) and an organ mapping antibody panel (OMAP) was developed and published as part of the 4th HRA release[29]. We used two cytokeratin cocktails CK26 (KRT1, KRT5, KRT6, and KRT8) and AE1 (KRT10, KRT14, KRT15, KRT16, and KRT19), which had broad keratinocyte coverage. For example, PCK26 stained the entire epidermis and AE1 was more localized to the lower half of the epidermis, highlighting the stratum basale. Both cytokeratin cocktails also stained adnexal glandular and hair follicular units in the dermis. However, additional staining with specific keratins would be required to resolve each of the specific cell types (e.g., KRT10 for granular keratinocytes[30]). In future studies, additional markers can be included to achieve greater cell specificity (e.g., hair follicle stem cells) and combined with spatial transcriptomic methods to achieve even higher resolution.

**Hybrid supervised and unsupervised approach for accurate cell segmentation and classification.** The tissue block containing the 12 skin samples first underwent micro CT imaging (see Methods and Supplementary Fig. 4), followed by sectioning into $100 \times 5$ μm sections. Twenty-six of the best-quality serial sections were downselected for analysis (to minimize spacing, no more than two sequential sections were excluded between each retained section). Each section underwent multiplexed immuno-fluorescence imaging with 18 skin biomarkers plus nuclear marker DAPI (see Methods). Our method provides an integrated workflow for 2D segmentation and classification of cell types from the multiplexed images, followed by automated 3D reconstruction (see Methods and Supplementary Fig. 5a, b). Cell type classification is not usually integrated into segmentation work-flows, and manual thresholding/gating of biomarker signal or clustering of segmented cells is often used, which is manual and prone to errors. We developed a hybrid supervised and unsupervised segmentation/classification model where a supervised deep learning (DL) model was first used for 2D DAPI/nuclei segmentation, followed by unsupervised Gaussian mixture models (GMM)[31] for probabilistic segmentation/classification of individual cell-type (i.e., epithelial and immune) and DNA damage/repair and proliferation markers (i.e., p53, DDB2, and Ki67). GMM is an excellent tool for simultaneous image normalization and detecting relative changes in biomarker intensity, allowing robust classification in each section. Combining DL and GMM provides a generalizable solution for cell segmentation and classification that works for large datasets of whole slide images. While there are several open source options available for cell segmentation (e.g., CellSeg[32], Cell Profiler[33], or StarDist3D[34]), these would have been relatively time-consuming to implement here given the large amount of data generated for 26 serial sections (~15 GB and ~40 stitched FOV per sample ($0.832$ mm $\times 0.702$ mm/FOV). Typically, thousands of manually annotated cells are required to develop a DL-based segmentation model, and manual annotation introduces inter- and intra-rater variability. For example, the development of the CellSeg model required 29,000+ manually segmented nuclei to build a DAPI-based nuclei segmentation model[32]. Our hybrid approach is faster and more generalizable for handling larger tissue images, and it does not rely on the manual tuning of image thresholding values, image normalization, and morphological operations (median filtering, difference of Gaussian) and watershed algorithm parameters (such as gradient thresholding and diffusion values). For the purpose of this study, it is currently limited to nuclear segmentation and nuclear/peri-nuclear markers, but the workflow is adaptable and can be expanded to whole cells through dilation or including membrane markers such as pan-cadherin or Na+K +ATPase[35]. We found excellent sensitivity (93–100%), specificity (85–100%) and accuracy was above 90% for all markers (Fig. 2a–e), compared to manual annotations (see Methods). We further benchmarked the immune cell counts against previous literature in healthy skin and found comparable results. After scaling each sample to 1 cm × 5 μm, we estimated a median count of 712 T cells (SD = 329, $n = 10$), which is comparable to the previously reported median of 590 T cells (SD = 105) in 1 cm × 5 μm skin sections[36]. More than 95% of T cells in normal skin have been identified as T memory or helper cells[37] with T reg cells estimated to be 10% of the total T cell population[38]. We also found a similar distribution of T helper (89 ± 5%); T killer (2 ± 5%), and T regs (9 ± 5%).

**3D reconstruction of multiplexed serial sections.** For 3D reconstruction of the serial sections, we first manually select one reference 2D AF image from the stack of 26 serial images (Fig. 3a, b and Methods). All AF images used for reconstruction are unstained (i.e., the image is taken prior to any marker staining), hence AF-based registration is independent of biomarker signal or signal variations associated with staining. Compared to other registration approaches that have been used for 3D reconstruction[14], we automatically segment AF in each serial image using Otsu thresholding, morphological closing, and retaining the largest component[39] (Fig. 3c) to generate a 2D AF mask. The 2D AF mask focuses registration on a region of interest (ROI) and filters out background noise or artefacts that may interfere with the registration process. Post serial image registration, 3D volumes of AF and cells are created, and 3D connectivity of all cells is used to fuse overlapping cells in adjacent serial sections to prevent overcounting (Fig. 3d, e). We use ITK's 3D connected component image filter to merge overlapping cells and classify cells in 3D[40,41]. Connected component image filter has historically been used in merging segmentations in 3D[42–44] and for refining cell segmentation in 2D[45]. To the best of our knowledge, this is the first time a 3D connected component has been used to fuse overlapping cells in 3D in serial histological sections. Mean dice similarity coefficient (DSC)[46] was used to assess image registration accuracy, and we found an overlap accuracy of 0.95 ± 0.04 for 24 serial sections in the ten reconstructed skin volumes (the last two sections were used for deep learning training and excluded). An overlap DSC of 0.90 is classified as high quality in the 3D registration or reconstruction community[14]. The normalized cross correlation between the serial AF sections (a metric of registration quality) was 0.6 ± 0.07 for all AF serial sections. This result is comparable to established 3D reconstruction methods[13] and is good considering the deformation/damage that can occur during cyclic multiplexing, which could negatively affect accurate registration. Slides were also randomly picked from each of the reconstructed skin volumes and segmented cells and biomarkers were overlaid on the images for visual validation. To mitigate potential issues associated with higher or lower signal intensity for a slide, any discrepancies in cell density of a biomarker were identified and a higher or lower probability threshold was used on the probabilistic segmentation to improve segmentation. Such manual biomarker probability adjustments were necessary in less than 2% of the whole slide images dataset.

**Interactive visualization and quantification of 3D cell density and distances.** Understanding and communicating the 3D spatial location and distance relationships of multiple cell types and supporting the comparison of cell type distance distributions across donors and conditions is non-trivial. For this study, three interactive visualizations were developed to serve this need: (1) cluster density plots that illustrate and quantify the number of cells within a specified distance (we evaluated both 15 and 30 μm, with the latter providing more biologically meaningful results); (2) a 3D vasculature CCF visualization (VCCF) to illustrate the spatial relationships and distances between immune cells and nearest endothelial cells in 2D and 3D; and (3) spatial illustrations that quantify the distances between UV-damaged and proliferating keratinocytes and the skin surface. The interactive 3D visualizations are accessible via the companion website: Companion Website for "Human Digital Twin: 3D Atlas Reconstruction of Skin and Spatial Mapping of Immune Cell Density, Vascular Distance and Effects of Sun Exposure and Aging" | (hubmapconsortium.github.io)) and make it possible to examine one 3D reconstructed region at a time. Users can view one or more serial sections; they can view one or more cell types/ markers in these sections; they can review the automatically updated distance distribution plots below the 3D skin visualization; plus, they can access the virtual H&E image (a mid-point

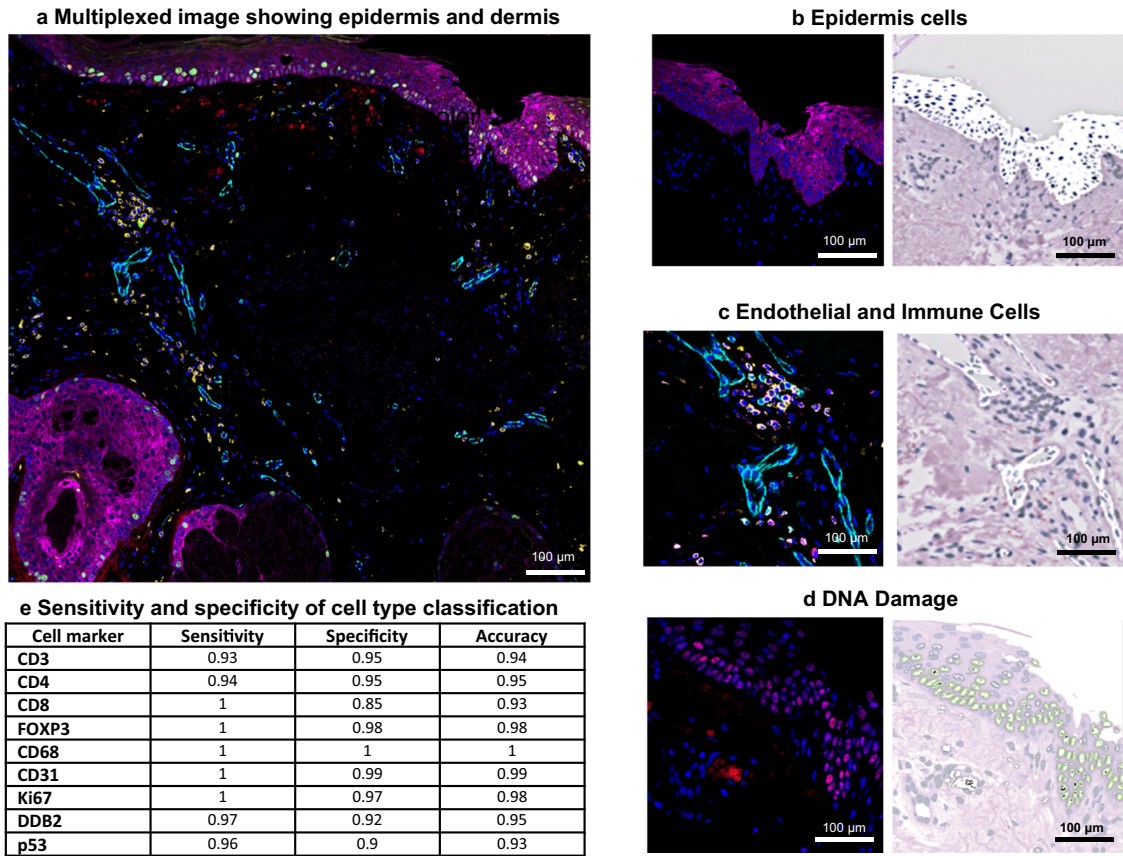

**a Multiplexed image showing epidermis and dermis**

**b Epidermis cells**

**c Endothelial and Immune Cells**

**d DNA Damage**

**e Sensitivity and specificity of cell type classification**

| Cell marker | Sensitivity | Specificity | Accuracy |
|---|---|---|---|
| CD3 | 0.93 | 0.95 | 0.94 |
| CD4 | 0.94 | 0.95 | 0.95 |
| CD8 | 1 | 0.85 | 0.93 |
| FOXP3 | 1 | 0.98 | 0.98 |
| CD68 | 1 | 1 | 1 |
| CD31 | 1 | 0.99 | 0.99 |
| Ki67 | 1 | 0.97 | 0.98 |
| DDB2 | 0.97 | 0.92 | 0.95 |
| p53 | 0.96 | 0.9 | 0.93 |

**Fig. 2 Multiplexed images of cell type markers and GMM probability maps overlaid on virtual H&E. a** Exampleregion of interest within a multiplexed image from Donor 9 (region 3); GMM was used to automatically segment/classify different cell types. Sub-regions of interest are shown in **b**–**d** panels with probability maps for each cell type; **b** cytokeratin CK26 epidermis staining and adjacent probability map of epidermis cells overlaid on the virtual H&E; **c** endothelial cells of blood vessels (CD31) and adjacent probability map of endothelial cells; **d** p53 staining in the epidermis and adjacent probability map of p53 positive cells; **e** The sensitivity, specificity, and accuracy metrics for the cell classification workflow for nine biomarkers using manual annotations.

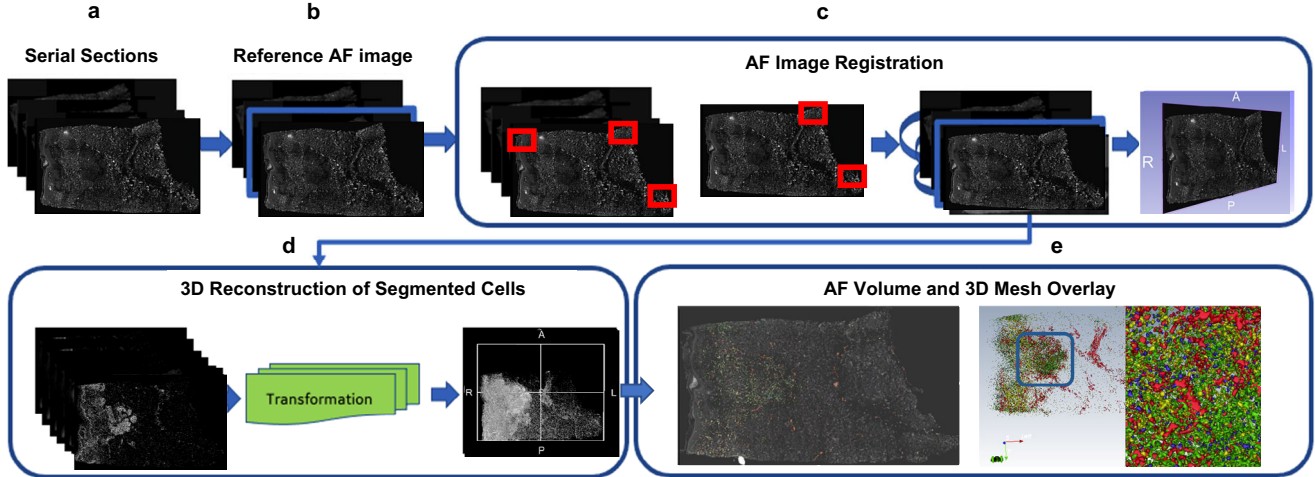

**Fig. 3 3D skin volume reconstruction of autofluorescence skin images and segmented cells. a**, **b** A reference autofluorescence image from the sequentially imaged sections is used to initiate registration; **c** A patch-based local correspondences in 2D serial sections is used for affine registration followed by deformable registration to account for tissue deformation; **d** 3D reconstruction of the segmented cells is achieved by mapping the cells in 3D using the affine and deformable transformation map and refinement is achieved by registration to micro CT image; **e** 3D volumes of classified cells overlaid on the AF volume and 3D mesh.

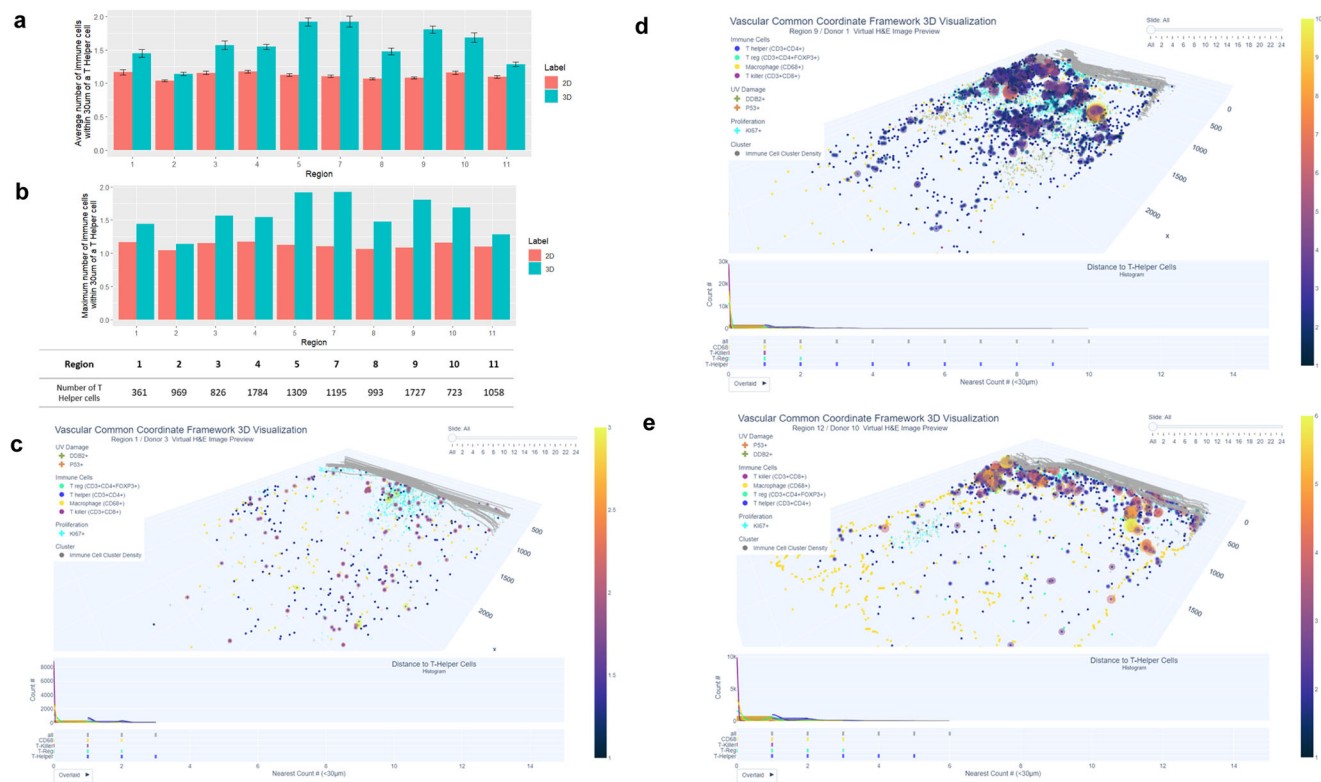

**Fig. 4 Higher spatial density of T cells in 3D vs 2D and visualization of immune cell cluster density. a** Average number of all immune cells within 30 μm of a T helper cell in 2D (red) and 3D (blue) are shown for each donor sample. Overall, the number of immune cells within 30 um of a T helper cell was higher in 3D compared to 2D with 10–70% more cells found in 3D. Error bars represent the 95% confidence interval. Each error bar was derived from independent T helper cells detected in each region ($n = 10$ biologically independent samples). The number of T helper cells for each region is also shown; **b** Maximum number of immune cells within 30 μm of a T helper cell. There was variation across all regions, for example, in region 7 there a cluster of 11 immune cells within 30 μm were found in 3D while three cells were quantified in 2D. **c** Visualization of low immune cell cluster density in 3D. Immune cell cluster density plot is generated from region 1 (scalp region, marked sun exposure, male, 72 years of age). The size and color of the purple bubbles show that this sample has a relatively low number of neighboring immune cells in 3D. For interactive visualization of 3D and 2D sections go to: https://hubmapconsortium.github.io/vccf-visualization-release/html/immune_cluster/immune_cluster_region_1_30.html **d** Visualization of medium-high immune cell cluster density in 3D. Medium-high immune cluster density plot from region 9 (lower distal arm region, marked sun exposure, male, 60 years of age). The increased number and size and colors (going from pink to yellow) of the bubbles sample show the higher number of neighboring immune cells in 3D. For interactive visualization of 3D and 2D section: https://hubmapconsortium.github.io/vccf-visualization-release/html/immune_cluster/immune_cluster_region_9_30.html **e** Visualization of high immune cell cluster density in 3D. High immune cluster density plot from region 12 (biopsy was taken from the right flank but excluded from further analysis due to scarring; the donor also had systemic lupus erythematosus). The high number of orange and yellow bubbles shows a high number of neighboring immune cells in 3D (4–6+ cells with the highest density clusters, according to the color-coded legend) and demonstrates high immune cluster density. For interactive visualization go to: https://hubmapconsortium.github.io/vccf-visualization-release/html/immune_cluster/immune_cluster_region_12_30.html. All regions can be accessed at: Companion Website for "Human Digital Twin: 3D Atlas Reconstruction of Skin and Spatial Mapping of Immune Cell Density, Vascular Distance and Effects of Sun Exposure and Aging" | (hubmapconsortium.github.io).

image in the 26 image stack was selected as a representative image) for histological context. As described in more detail below, static examples are provided in Fig. 4c–e (immune cluster density), Fig. 5c (distance between immune cells and endothelial cells); and Fig. 6b, c (distance of keratinocytes positive for p53 (DNA damage), DDB2 (DNA repair), and Ki67 (proliferation) to the skin surface).

**T cell counts in 3D skin volumes.** Previous measurements of total T cell counts in the skin have been based on 2D tissue sections[36]. Using the 3D reconstructed volumes and after correction for overlapping cells, we estimated the median number of T cells/cm³ skin to be $33.5 \times 10^6$ (SD $15.6 \times 10^6$) (Supplementary Table 4). This result provides further evidence that skin is a vast reserve of T cells, and the wide variation is attributed to anatomical variations in skin thickness, aging, sun exposure, health status, etc. Since skin thickness is not uniform, an ideal study

design would collect and analyze samples from different anatomical sites in the same patient, which was not possible here. There were no significant differences in normalized counts (adjusted for tissue 3D volume) in macrophages, T killer cells, T helper cells, or T reg cells by age or in donors' skin with mild to marked spectrum chronic sun exposure-related changes (Supplementary Fig. 6a–g). There was a trend for a positive relationship between the T helper/T killer ratio and age (corr = 0.82, adj. $p = 0.07$) (Supplementary Fig. S6h).

**T cell density is higher in 3D vs. 2D.** We then compared immune cell density in 2D vs 3D as the average number of T cells within 30 μm of a T helper cell (Fig. 4a) and the maximum number of T cells within 30 μm of a T helper cell (Fig. 4b). There was wide variation in both measurements across all samples due to heterogeneous cell density in each section/sample. Overall, we quantified 10–70% more T cells within 30 μm of a T helper cell in

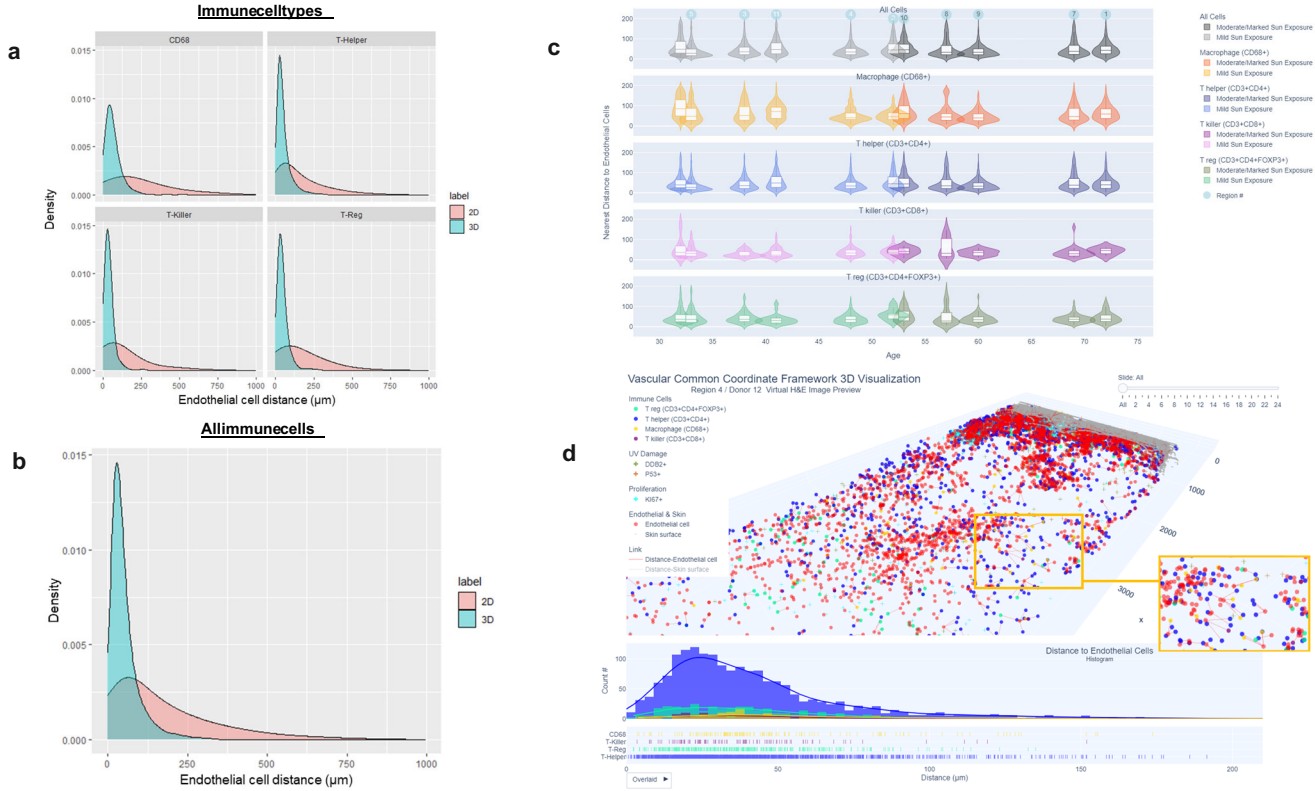

**Fig. 5 Distances between immune cells and endothelial cells in 3D vs 2D. a** Histograms of distance to nearest endothelial cells from each immune cell type (CD68, T Helper, T Killer, and T Reg). 3D distance to immune cells is typically much shorter (on average, ~56 µm in 3D vs 108 µm in 2D); **b** Two sample Kolmogorov–Smirnov test (two-sided) was performed to confirm that there is statistically significant difference between the 2D and 3D distributions for all immune cell types ($D = 0.43$, p value <2.2e-16; $n = 13,481$ (in 2D) and $n = 13,489$ (in 3D) independent immune cells were included in each distribution). **c** Distance between immune cells and nearest endothelial cells is consistent with aging and sun exposure. Violin plots are shown for each donor and sorted by age. There was a trend for a higher number of T killer cells closer to endothelial cells in younger vs older patients (spearman correlation = −0.73 ($p = 0.02$, adjusted p value = 0.08, Supplementary Fig. 7c ($n = 9$ independent samples were used). The interactive version of this plot is located at: https://hubmapconsortium.github.io/vccf-visualization-release/html/violin_cell.html **d** Example of skin region with a higher number of immune cells within 100 µm of endothelial cells. Reconstructed 3D distance map for immune cells and nearest endothelial cells. Example shown is for region 4 (superior abdomen, mild sun exposure, male, 48 years of age); histogram plot showing the distribution of immune cells within 50–200 µm, with the highest T killer cell count within 25–50 µm. For interactive visualization in 3D and 2D go to: https://hubmapconsortium.github.io/vccf-visualization-release/html/region_4.html).

3D vs 2D. For example, regions 1 and 2 had similar maximum number of immune cells ($n = 3$) within 30 µm of a T helper cell in 2D and 3D, whereas region 7 had a maximum of 11T cells in 3D, and just 3T cells in 2D. This difference is important for samples with low immune cell density, where analysis of spatial relationships and cell–cell interactions in 2D would be more challenging to accurately quantify. The immune cell cluster density plots in Fig. 4c–e illustrate three contrasting examples of skin regions with low (region 1), medium-high (region 9) and high (region 12) immune cell cluster density in 3D, respectively. The low and high examples may be attributed to the health/therapy status of the donors who were noted as having rheumatoid arthritis (region 1) and systemic lupus erythematosus (region 12).

**Shorter distances between immune cells and endothelial cells in 3D vs. 2D.** Constructing a vasculature-based coordinate system makes sense biologically as almost every living cell must be within a small distance to a blood vessel (~25 µm to 1 mm, depending on the tissue) to receive oxygen[47]. Aging has been shown to reduce the size and density of blood and lymphatic vessels in the skin as well as disrupt its structure[48]. We found no significant differences in endothelial cell numbers, regardless of age or sun exposure. There were significant differences between

2D and 3D in the average distance of the nearest endothelial cell to immune cells (macrophages, T helper, T killer, and T regs), and distances were typically shorter in 3D (~56 µm in 3D vs 108 µm in 2D, ($p < 0.0001$) (Fig. 5a, b). Distances between each immune cell type and endothelial cells in 3D are also shown as violin plots and grouped by age and sun exposure for each donor/region in Fig. 5c. There was a trend for higher counts of T killer cells within 100 µm of endothelial cells in younger donors (corr = −0.73, adjusted p value = 0.08; see Supplementary Fig. 7e). The implications of this are unclear without further validation in a larger group of subjects but may reflect age-related differences in adaptive immune response. An example of a region with higher total immune cell counts (including T killer) within 100 µm of endothelial cells is shown in Fig. 5d.

**No differences in spatial location of sun damage/proliferation cell markers age or sun exposure.** The total count of p53, Ki67, and DDB2 positive keratinocytes in 3D volume and distance to the skin surface was also calculated and distributions are shown as violin plots (Fig. 6a). The premise for this analysis is that higher counts of p53 damaged and/or Ki67 positive proliferating keratinocytes in the upper epidermis, or towards the skin surface, are an indicator for early pre-cancerous lesions. We quantified

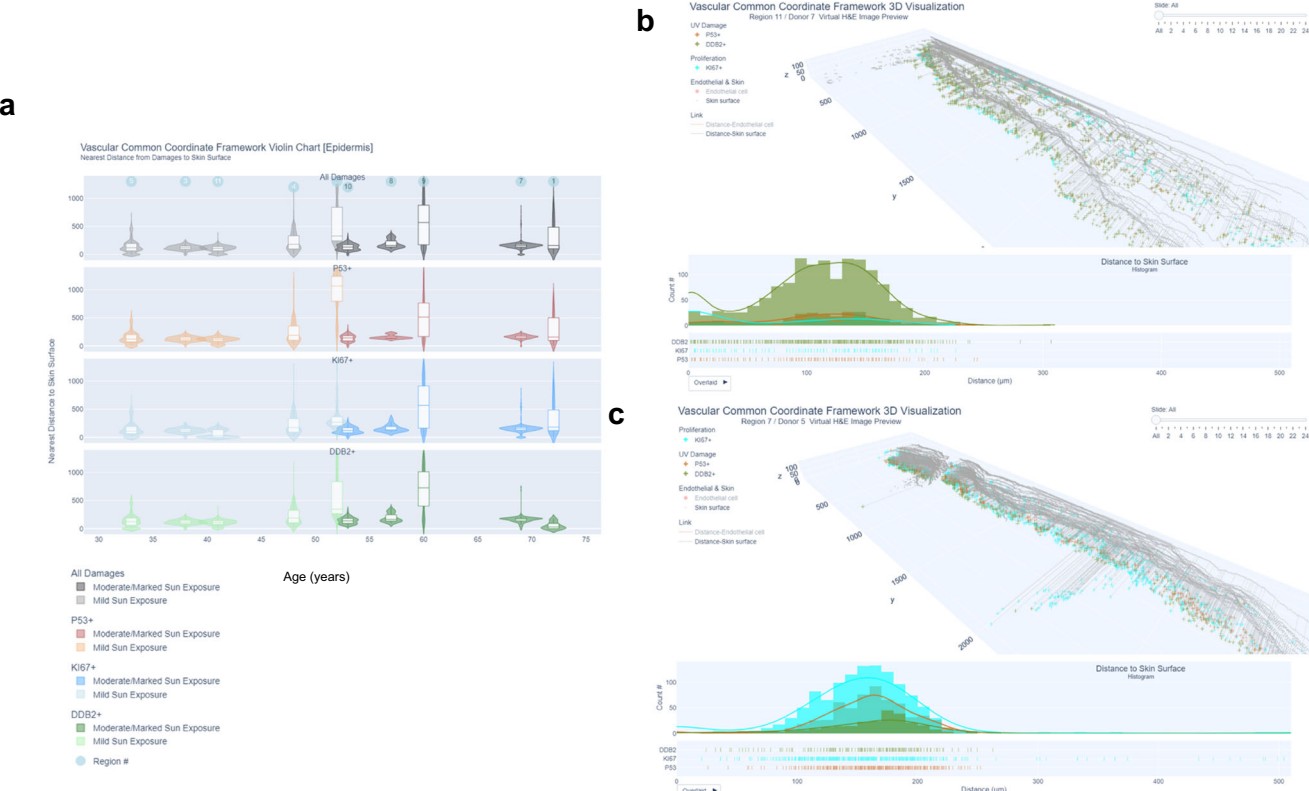

**Fig. 6 Distances between DDB2, p53, and Ki67 positive cells and the skin surface were consistent with age and sun exposure, and younger donors had higher counts of DDB2 positive cells. a** Violin plots for distances between DDB2, p53, Ki67 positive cells, and skin surface are shown for each donor and sorted by age. Most positive cells were localized within a 100–200 μm distance from the skin surface. However, regions 1, 9, and 2 had positive cells up to 1600 μm, which were localized to hair follicles. The interactive version of this plot is located at: https://hubmapconsortium.github.io/vccf-visualization-release/html/epidermis_entire/violin_damage_epidermis.html **b** Example of higher distribution of DDB2 positive cells within 200 μm distance of the skin surface in a younger donor. While we did not find differences in distances to the skin surface in relation to aging or sun exposure, there was a significant inverse correlation between DDB2 positive cells and age (corr = −0.78, adj. *p* = 0.05, Supplementary Fig. 9a), suggesting the higher capacity for DNA repair in younger donors. The example is from region 11 (from the upper arm, mild sun exposure, female, 41 years of age) and shows a higher distribution of DDB2 positive cells within 200 μm distance from the skin surface and lower distribution of cells positive for p53 and Ki67. For interactive visualization go to: https://hubmapconsortium.github.io/vccf-visualization-release/html/epidermis/epidermis_region_11.html **c** Example of lower distribution of DDB2 positive cells within 200 μm distance of the skin surface in an older donor. The example shown is region 7 (from the lower forearm, marked sun exposure, male, 69 years of age) and shows the highest distribution of Ki67 positive cells within 200 μm distance from the skin surface, with lower p53 and lowest DDB2 positive cells. For interactive visualization go to: https://hubmapconsortium.github.io/vccf-visualization-release/html/epidermis/epidermis_region_7.html. All regions are at: Companion Website for "Human Digital Twin: 3D Atlas Reconstruction of Skin and Spatial Mapping of Immune Cell Density, Vascular Distance and Effects of Sun Exposure and Aging" | (hubmapconsortium.github.io).

distances of p53, Ki67 and DDB2 positive keratinocytes to the skin surface using two different epidermis masks: (1) using AE1 cytokeratin cocktail, which was more specific for the lower epidermis/stratum basal layer and hair follicular units; and (2) CK26 cocktail, which stained the entire epidermis, as well as hair follicular units. Due to the non-uniformity of the skin surface, the spatial cell distance analysis was conducted using a hybrid of 3D and 2D data, whereby the distances of the 3D reconstructed cells to the skin surface were calculated using the nearest 2D tissue section. We found that most Ki67 and p53 positive keratinocytes were largely localized to the AE1+/stratum basal region (where regenerating keratinocyte stem cells are localized[49]). There were no significant differences in the distance of p53, DDB2, and Ki67 positive keratinocytes to the skin surface when analyzed by sun exposure or aging. Notably, there were three cases (regions 1, 2, and 9) that had a very wide spatial distribution of p53, Ki67 positive cells (up to 1600 μm from the skin surface (Supplementary Fig. 8a–c). In each case, this was due to a hair follicular unit extending deeper into the dermis with a high number of p53 and Ki67 positive cells. The number of p53 positive cells has been

shown to extend deeper into the hair follicles and glands in older patients[50] (these samples were from donors aged 52–72 years, however, we did not have matched younger patients for comparison). In all other cases, the average distance of p53 and Ki67 cells from the skin surface was 155 and 143 μm, respectively. Total Ki67 and p53 positive cell count was not significantly correlated with age or sun exposure (Supplementary Fig. 9). This agrees with previous studies that show that Ki67 and/or p53 positive cells do not increase until actinic keratosis (damage to keratinocytes in the lower third of the epidermal layer) or a precancerous/cancerous (full epidermis thickness) *squamous* in situ state is reached[22]. There was a significant inverse relationship between DDB2 positive cells and age in the stratum basale region of the epidermis (corr = −0.78, adj. *p* = 0.05, (Supplementary Fig. 9e) and the entire epidermis (corr = −0.79 adj. *p* = 0.04) Supplementary Fig. 9j). Given the critical role of DDB2 in nucleotide excision repair[51], this suggests a decreased capacity for DNA repair with aging. Figure 6b, c illustrates cell data from two example regions from younger and older donors with mild and marked sun exposure, respectively. Notably, the younger donor

has a higher number of DDB2 positive cells distributed through 0–200 μm of the epidermis and relatively low counts of p53 and Ki67 positive cells, compared to the older donor who has nearly the opposite profile (high Ki67 and p53 and low DDB2 positive cells).

## Discussion

We have demonstrated a workflow for spatially registering multiplexed tissue data in three dimensions using the HuBMAP registration user interface[23]. Interactive tools allow visualization of spatial patterns of cell types and distances in relation to vasculature and position within the epidermis. We have demonstrated higher immune cell density and shorter distances of major immune cells to the nearest blood vessel in 3D, in support of a vasculature-based human common coordinate framework[47]. Antibodies were aligned with the skin anatomical structures, cell types, and biomarker (ASCT + B) tables in support of high quality, ontology-aligned data generation. While these findings are in healthy skin, this 3D workflow is extendable to other organs, cell types, and disease states and thus provides a standardized approach for cellular resolution 3D spatial analysis and for constructing a human reference system. All datasets and code for segmentation, classification, reconstruction, and data visualization are freely available at GitHub - hubmapconsortium/ MATRICS-A: Multiplexed Image Three-D Reconstruction and Integrated Cell Spatial -Analysis and via a HuBMAP collection at Samples | HuBMAP (hubmapconsortium.org).

There are several experimental design factors to consider when planning a 3D reconstruction of serial sections using multiplexed immunofluorescence images. For the greatest efficiency (cost/ time), we embedded 12 samples in a single paraffin block, which also allowed consistent staining and imaging across all samples. Although there is some trade-off in terms of reduced cellular heterogeneity (compared to larger whole slide sections), this was offset by having a diverse range of samples from a wide age range and sun exposure effects and anatomical location. The blocks were cut into $100 \times 5$ μm sections and 26 of the highest quality sections were downselected based on correct placement on the slide and no visual imperfections such as tears or missing tissue. This is important for multiplexed imaging processes where the coverslip is removed between each staining round and may result in damage to the tissue (in our experience, if tissue loss occurs, it typically takes place in the first 1–3 rounds of staining, and especially if samples have small tears due to fragile nature or sectioning). The use of Superfrost™ slides is an important mitigation against tissue loss, due to stronger adherence of tissue to the positively charged slide surface. More recent commercial options for multiplexing include the use of a flow cell or coverslip-free format, hence tissue loss may be less of a concern for future work. Serial sections were all stained in a single batch with well-characterized antibodies, providing high quality, consistent images. Quenching methods are sometimes used to reduce signal in highly autofluorescent tissues (e.g., Sudan black B treatment has been reported to reduce AF signal by 60–95% in pancreatic tissue[52]), however, we did not use quenching to reduce AF (background/AF is imaged separately and subtracted from true biomarker signal) and further evaluation is needed to determine if quenching would negatively affect our AF-based registration workflow. Although there is inherent variation in AF signal from section to section or from sample to sample, we address this using the Otsu thresholding approach, which uses intra-class variance to set the AF threshold for each sample/slide. The AF threshold is automatically adjusted from one serial section to another to maximize the separation of the background and the foreground.

As described earlier, one limitation of the 3D reconstruction of tissue sections has been the introduction of the "banana effect" which erroneously straightens curved anatomical structures during registration. Micro CT imaging of the tissue block prior to sectioning, provides a reference volume to which the 3D reconstructed volume of the skin tissue and the cells are registered. A similar approach has also been successfully applied for whole human brains using MRI, blockface photography (to bridge the gap between MRI and histology), and thick sequential histological brain sections (25 μm thick)[17]. Posterior registration/alignment of the 3D reconstructed volumes to micro CT images is valuable when there is deformation and/or wear and tear in the tissue samples during the cyclic staining process. The co-registration maps microfeatures (e.g., cell types) to macro imaging features, and compared to landmark-constrained 3D histological imaging[53], minimal manual intervention is required for accurate registration and reconstruction of the 3D volume. Within our workflow, a single slide from the entire volume is identified manually based on tissue quality as the reference image; the rest of the process for registration and 3D reconstruction is completely automatic. Compared to image similarity-based alignment[14,41], automatic block correspondences is used for initial affine alignment, which improves registration speed[39,41] and also accounts for local deformations that may happen during the staining and image acquisition process.

MATRICS-A was evaluated in skin specimens sampled across various body locations to account for diversity in an anatomical organization and differences in sun exposure. Combining 3D reconstructed cellular data with interactive visualization provided several insights that have not previously been shown using either 2D or 3D volume imaging of skin by confocal or LSFM. We show that that T cell counts in the 3D reconstructed volumes ranged from 500–2000 cells and scaled to an estimated 15–60 million in 1 $cm^3$. To our knowledge, the only prior work to quantify the pool of T cells in the skin was done in 2D by ref. [36] where they estimated 20 billion T cells in 1.8 $m^2$ (average surface area of skin). One important consideration when interpreting volumetric immune cell counts is the anatomical location of the skin sample which affects the keratin layer thickness, as well as the distribution and density of adnexal structures such as hair follicles, and cumulative sun exposure. Partially addressing this, our samples were collected from across the anatomy, including arms, legs, abdomen, and scalp, and normalized for volume and endothelial cell count to account for sample-to-sample differences. However, an ideal study design would be to prospectively collect at least two distinct anatomical samples from the same donor and expand racial diversity, which is planned as part of future efforts on HuBMAP. While our 3D reconstructed skin volumes are relatively small (we used $100 \times 5$ μm sections and downselected $26 \times 5$ μm sections), this is on a par with confocal imaging where typical imaging depth is 50–100 μm (up to 2 mm is possible with customized clearing protocols[54]), but with limited multiplexing capability (4–5 markers). Light-sheet fluorescence microscopy (LSFM), coupled with clearing protocols, achieves the highest imaging depth (between 1 $mm^3$ and 1 $cm^3$) and up to 7 $cm^3$ prostate and brain samples was recently reported[55]. However, the maximum number of markers still remains relatively low, compared to the multiplexing potential in thinner tissue sections (e.g., LSFM imaging of 9 proteins in 500 μm thick cleared brain sections was recently reported[56]). It is also constrained by long antibody incubation times to achieve maximum and uniform staining penetration. We demonstrated our reconstruction workflow with 26 sections, but a larger number of multiplexed sections is technically feasible and is limited more by cost than technical constraints. Other factors to consider in 3D analysis include imaging resolution, including the axial resolution

(Z-dimension) which is typically 500–700 nm and 200–300 nm lateral resolution (X-Y dimensions) at 20x magnification in standard optical microscopes[57]. Label free 3D imaging methods can provide anatomical or molecular information but can have trade-offs in resolution. For example optical coherence microscopy provides a large field of view, but has lower resolution (1–15 μm); or Raman microscopy provides molecular information but resolution is diffraction limited and can require stronger excitation lasers to increase molecular signal relative to the background fluorescence and spatial resolution[57].

Even with all the recent advancements in 3D imaging, only a limited number of open access analysis tools are available[58], hindering progress in the interpretation of 3D data. We provide visualization tools for 3D reconstructed cellular data including spatial cell distance analysis and 3D cell cluster density. These provide unique representations of 3D cellular data and future iterations could be combined with H&E images to increase the depth and provide more anatomical context. Using conventional bar-charts we demonstrated 10–70% more T cells within 30 μm of a T helper cell in 3D vs 2D, however cluster density plots more clearly visualized T cell density variations in 3D. These plots are inspired by 3D geospatial and population density maps[59] and combine scatter plots and heatmap in a 3D view. The combination of these plots allows us to focus on the distribution pattern of each individual immune cell while still observing the overall distribution. We anticipate that 3D density plots (and versions thereof) will become increasingly important for the 3D microscopy field in healthy, aging, and diseased tissue. We found significant differences in the average distance of the nearest endothelial cell to immune cells, with distances in 3D half that found in 2D (~56 μm vs 108 μm on average). Using confocal imaging, ref. [18] showed that T cells form perivascular sheaths throughout the dermis and reside within 15 μm distance from endothelial cells (in breast skin). Indeed, we found that most T cells were located within 25 μm of nearest endothelial cell across all ages, but we also see a wider range of distances up to 150 μm. p53 and Ki67 positive keratinocytes were largely localized to the stratum basale where the stem cells are located, but distances to the skin surface or counts did not vary by aging or sun exposure. Collectively these spatial metrics deepen our understanding of the interactions between immune cells and proximity to vasculature and effects of aging/sun exposure and opens opportunities for further applications in disease context and other organs. Additional insights will be gained from incorporation of additional markers of aging, senescence, immune (e.g., Langerhans cells), functional immune markers (e.g., exhaustion or activation[60]) and inflammation markers. Integration of scRNAseq and spatial transcriptomic data from the same samples will further dissect the complexity of skin cell populations and will require prospective sample collection and different protocols for sample preservation (which is planned as part of ongoing HuBMAP organ projects, including skin).

While 2D spatial analysis will continue to be the method of choice for most researchers' due accessibility and cost, it is important to consider what histological or anatomical information is being minimized or lost in thinner tissue sections, and plan sample collection in accordance with the question/s being asked. As recently shown by ref. [12], 2D whole slide images may be sufficient for quantifying the range of cellular interactions and neighborhoods within a tumor or organ (and superior to small tissue cores), but to understand cellular relationships in relation to anatomical structure and tumor budding, reconstructed 3D images provided more contextual information. Future solutions could combine higher dimensional 3D microscopy (confocal/lightspeed) with sequential sectioning from adjacent tissue regions. This may require prospective tissue collection with careful selection of regions of interest for each modality and advance planning sample preservation. Due to the cost/time to do these methods and the computational requirements for analyzing the large volumes of data there will have to be some trade-offs e.g., 3D analysis could be limited to smaller, well defined regions of interest within an organ and combined with 2D whole slide data. HuBMAP and the other reference mapping efforts are providing a unique opportunity to conduct 2D and 3D spatial cellular analysis across many organs, using standardized methods for tissue imaging, including spatial protein and transcriptomics analysis, combined with non-spatial methods such as scRNAseq. This will lay the foundations for deepening our understanding of cell types and spatial relationships across the human body in 2D and 3D.

## Methods

**Patient samples**. Skin biopsies were collected from 12 donors ranging from 32–72 years with a mix of typically UV-exposed and non-exposed anatomical regions (Supplementary Table 1). The biopsies were trimmed and embedded in a single block that underwent micro CT imaging. The blocks were then sectioned into $100 \times 5$ μm serial sections, numbered in sequence, of which up to 26 of the highest quality in serial succession were selected for further analysis (slide layout shown with virtual H&Es –comprised of pseudo-colored autofluorescence and DAPI[61] in Supplementary Fig. 1). All 12 biopsies were spatially registered using the HuBMAP Registration User Interface along with their corresponding metadata, including donor information, including health status and sample processing (https://hubmapconsortium.github.io/ccf-ui/rui/ - Supplementary Fig. 2). Of the 12 samples, ten were downselected for further statistical analysis. The two excluded samples included a donor with a benign cyst, but with extensive inflammation and immune cell infiltration compared to other samples (region 6). The second sample had a scar which also altered the normal organization of the epidermis and dermis layers (region 12). All donors were in good health and cancer free at the time of sample collection, with two donors having chronic diseases (RA and HIV), which is noted in the patient summary table (Supplementary Table 1). All patients consented to provide samples for research and relevant ethical regulations were followed. The study protocol was approved by University of Pittsburgh IRB (STUDY19120023).

**Pathologist review**. Virtual H&E images (which are pseudo-colored autofluorescence and DAPI images[61]) from each donor were assessed by a pathologist for histopathological changes related to chronic sun exposure such as keratinocytic atypia in the epidermis and degree of solar elastosis changes in the dermis. Accordingly, the specimens were categorized into groups of skin with mild, moderate, and marked chronic sun exposure-related changes (Supplementary Table 1 and virtual H&E for two contrasting regions (mild vs. marked sun exposure) in Supplementary Fig. 10a, b). Donors in the mild chronic sun exposure group were significantly younger than the moderate-marked exposure donors (42.4 vs. 62.2 years, $p = 0.008$). All virtual H&E images are located at: vccf-visualization-release/vheimages at main · hubmapconsortium/vccf-visualization-release · GitHub

**Micro CT imaging of skin blocks**. A Phoenix micro CT system (GE, Wunstorf, Germany) with up to 300 kV/500 W was used to generate high-resolution CT images of the 12 skin samples within the tissue block. Example images for micro CT workflow with corresponding histological section are shown in Supplementary Fig. 4. Phoenix micro CT scanners have a high dynamic DXR digital detector array and can produce isotropic images of 1 μm and are frequently used for industrial process control as well as for scientific research applications. Due to its dual tube configuration, detailed 3D information for an extremely wide sample range can be provided. For our purpose a current of 200 kV was found to be optimum in terms of signal-to-noise ratio to generate high quality volumetric isotropic images of 0.016 mm resolution for the embedded skin samples and this also allowed imaging of 12 samples in one block within 30 min. The DICOM header with imaging settings is shown in Supplementary Table 5.

**Antibody validation**. All antibodies used in this study were subjected to a standardized characterization process which was developed by our lab and has been implemented for over a decade for over 500 commercial antibodies[28]. We typically start characterization using a reference multiorgan TMA (MTU391, Pantomics) which contains 15 major types of cancer (surgically resected) and corresponding uninvolved tissues as controls. The samples used for the reference TMA are typically fixed in 10% neutral buffered formalin for 24 h and processed using identical SOPs. To ensure consistent staining in new lots, clones or conjugates, all antibodies are re-tested on MTU391 arrays (depending on timing, these may be different donor samples but the same organ format). The MTU391 arrays are also used for optimizing concentration and testing whether the dye inactivation solution had any negative effects on the protein epitopes. Initial characterization and

down-selection includes (1) screening multiple clones/target that are compatible with immunohistochemical detection in FFPE tissue (using published literature and Human Protein Atlas[62]); (2) evaluating performance specificity using the MTU391 array and isotype control using a labeled secondary antibody; (3) to confirm epitope stability to the multiplexed cycling process, unstained MTU391 slides are processed through multiple rounds of signal inactivation and then stained to evaluate whether target intensity had decreased. In this study, none of the epitopes showed sensitivity to the signal inactivation protocol (i.e., staining intensity did not decrease following inactivation); (4) the best performing antibodies were conjugated to a fluorescent dye at multiple dye:protein ratios and titrated on sequential MTU391 TMA sections to compare sensitivity and specificity to the unmodified primary antibody. Primary secondary detection be also used in the first round of staining (assuming different species are available), which provides flexibility for any antibody that cannot be successfully conjugated. For the current study, all antibodies were tested in MTU391 arrays as described and then re-tested in a pilot study using ten skin samples provided by U. Pitt Dermatopathology department. Supplementary Table 3 shows the antibody clones and conjugates used in the study. The 18-marker panel provided coverage for 14 cell types: keratinocytes (granular, spinous, basal), epithelial, fibroblast, immune cells (macrophage, T helper, T killer, T reg), nerve, myoepithelial, and endothelial cells. These are also highlighted using the HuBMAP Anatomical Structures Cell Type and Biomarkers (ASCT + B) reporter comparison feature https://hubmapconsortium.github.io/ccf-releases/v1.0/docs/asct-b/skin.html. Examples shown in Supplementary Figure 11a–r depict representative staining on both the MTU391 tissue array and skin tissue. All antibodies showed good staining specificity in skin, with the exception of UCHL1 (nerve marker), which showed high background/non-specific staining, in addition to specific nerve staining (Supplementary Fig. 11r).

**Multiplexed imaging.** Multiplexed immunofluorescence (MxIF) staining and imaging of the skin samples was performed as described by refs. [61,63] using Cell DIVE™ technology (Leica Microsystems, Wetzlar, Germany). To summarize, slides are sectioned onto Superfrost™ (Thermo Fisher), which is critical for minimizing tissue loss during the cycling process. After slide clearing and using a two-step antigen retrieval process[63], the FFPE slides were stained with DAPI and imaged in all channels of interest to acquire background autofluorescence (AF) of the tissue. This was followed by primary/secondary and/or direct conjugate antibody staining of up to two markers per round plus DAPI, dye inactivation, and repeated cycle for all 18 biomarkers. Each sample had ~40 20x fields of view (FOV) and up to 26 sections underwent staining (using the Leica Bond) and imaging (IN Cell Imager 2200) in single batches to ensure consistency. A MTU391 slide (described above) was also included as a technical control. The Cell DIVE imaging software allows real-time image processing (registration, AF subtraction and illumination correction during each round of imaging). Multiplexed images were automatically registered and processed for autofluorescence (AF) subtraction[61,64] and illumination correction (described in more detail below).

**Illumination correction.** Since every skin specimen in the block was comprised of multiple FOVs (up to 40), this required image stitching over a large area. Inherent non-uniformity of illumination can give rise to "quilting" artifacts, in which the boundaries of single image fields appear distinct when combined with neighboring fields. To correct for this inherent non-uniformity, we performed several initial calibrations. In the first step, we cycle through all objectives/filters while imaging a standard blank glass slide. Focus is determined using a hardware laser autofocuser, which is part of the Cell DIVE imaging system. We acquire images of the blank glass at long exposure times, and subsequently use these images to subtract away any glints or reflections that might be present in the system. For the second step we image a series of fluorescent plastic slides (Ted Pella Fluorescence Reference, #2273). Again, we cycle through all objective/filter combinations, imaging the fluorescent slide closest to the emission peak of a given filter set. Before processing each of these images we subtract the appropriate image of background glass (i.e., for the same objective/filter and with intensity scaled by the ratio of exposure times). We are then left with an image that nominally should be uniform, but instead displays the non-uniformity of that filter or objective. We then utilize a fitting function to compute a set of parameters that characterize the non-uniformity (including determining the actual center of illumination). These parameters are saved, and for subsequent imaging we first subtract a properly scaled background glass image, and then divide out the non-uniformity using the pre-determined coefficients.

**Image normalization.** DAPI images are normalized to zero mean unit variance values inside our deep learning framework prior to nuclei segmentation. Secondly, the individual whole slides were normalized between zero mean and unit standard deviation before estimation of biomarker probabilities using GMM. Whole slide image specific normalization ensures all images were scaled relative to their intensity distribution and reduces intensity variability often observed between serial images. The combination of these two normalizations ensured a single cell was segmented based on relative intensity difference between the biomarker and the background and not on absolute intensity distribution that may vary from one slide to another adversely affecting segmentation accuracy.

**Cell segmentation and classification.** Figure 2 and Supplementary Fig. 5a, b summarizes our segmentation model framework. First, an encoder-decoder based deep learning (DL) model[65] was trained on a small sample (194 DAPI image patches selected from 30 images from ten patients) of manually annotated nuclei created using the annotation function in QuPATH[66]. Multiscale Laplacian of Gaussian (LOG) was introduced along with the DAPI images as separate channels to our encoder-decoder based DL model. The LOG feature detects blob like structures (which correspond to nuclei shape and boundary) in the DAPI images, and thereby provides contextual information to the DL model. Use of multiple channels allowed us to train an accurate DL model from a small sample of the manually annotated DAPI images. The depth of the encoder-decoder DL model was set to 4 and binary cross entropy was used as a loss function. An unsupervised GMM was then used for automatic probabilistic segmentation of immune cell types: T killer (CD8), T reg (FOXP3), T helper (CD4), macrophages (CD68), as well as endothelial cells (CD31), markers of proliferation (Ki67), DNA damage (p53), and DNA repair (DDB2) (Supplementary Fig. 5b). By design, GMM produces a probabilistic segmentation for cells and functional biomarkers, which are continuous probability values between 0–1 (i.e., 0.2, 0.3, 0.4, etc.). A threshold is then applied to create a binary classification. Since probability values are directly correlated to signal intensity, high expression would have probability value close to 0.9. GMM with two clusters was used to obtain a probabilistic segmentation of the cell-type/markers and background. Pixels with high signal for cell type and functional markers were automatically assigned high probability values in one cluster (positive class) and pixels with low signal intensity were assigned high probability values in the second cluster (background—negative class). Union of probabilities obtained from the positive class of GMM model and nuclei segmentation (from the DL model) were then fused. Probability values were used to automatically scale (between 0–1) and quantify positive cells (i.e., for each marker of interest) and determine the percentage overlap between the markers and the segmented nuclei. Low percentage overlap was further used to remove imaging artifacts, debris, and cells with low/background marker intensity. The same GMM was used to automatically segment contiguous structures, independent of nuclei, such as blood vessels (based on CD31 staining) and epithelial masks (based on AE1 and CK26 cytokeratin cocktail staining). In this scenario, we depend on probabilities as obtained from our GMM and automatically threshold the biomarker probabilities using Otsu filters[67]. All code can be found at GitHub - hubmapconsortium/MATRICS-A: Multiplexed Image Three-D Reconstruction and Integrated Cell Spatial -Analysis. The docker container/environment to run the code can obtained by using docker pull hubmap/gehc:skin and test data for skin region 7 can be found at Human Digital Twin: Automated Cell Type Distance Computation and 3D Atlas Construction in Multiplexed Skin Biopsies | Zenodo. There is a corresponding ReadMe file that provides context and instructions for the repository's contents and could be found here MATRICS-A/README.md at main · hubmapconsortium/MATRICS-A · GitHub.

**Manual annotations for cell classification accuracy.** For validation of the cell classification approach, a total of 2722 positive and negative cell markers were manually annotated using the annotation function in QuPATH[66]. The annotation breakdown is as follows - CD3: 408; CD4: 281; CD8: 347; FOXP3: 360; CD68: 391, CD31: 352; Ki67: 150; p53: 164; DDB2: 162. These datasets were then used to calculate cell classification sensitivity, specificity, and accuracy (Fig. 2e).

**3D reconstruction of multiplexed serial images.** For 3D reconstruction a reference AF image was manually identified from 24 serial sections of every region based on image quality (contrast, wear and tear, deformation etc.). All 2D AF images were masked (with a tissue mask) and all 2D AF serial section images were registered to the chosen reference image. Otsu thresholding uses intra-class variance computed from image histogram to set the thresholds for the AF background and the foreground. Hence while the AF signal distribution may change from one serial section to another, the threshold is set based on intra-class variance and is automatically adjusted from one serial section to another to maximize the separation of the background and the foreground. Our registration model uses local patch-based, normalized cross correlation to establish correspondences making our model fast without compromising on accuracy. Patch wise normalized cross correlation depends less on absolute intensity values and depends more on a good separation of the background and foreground signal which we obtain by masking our autofluorescence image. After affine registration, B-spline based deformable registration[68], we use normalized mutual information to mitigate image intensity differences that may occur between one serial section to another in the autofluorescence images. A block matching strategy[69] was adopted to determine the transformation parameters for the affine registration from masked AF images (Fig. 3c). The similarity between a block from the reference AF was computed relative to the AF serial sections. The best corresponding block defined the displacement vector for the affine transformation[70]. Normalized mutual information was chosen as the similarity matrix and maximized to achieve the registration. The transformation map obtained from the registration of the AF images was applied to individual biomarkers for all serial sections to create a 3D volume of endothelial, T killer, T reg, T helper cells and macrophages (Fig. 3d) and overlaid on 3D AF volume as shown in Fig. 3e. Post registration, 3D connectivity of all cells were used to fuse overlapping cells in adjacent serial sections to prevent

overcounting. Cells overlapping in 3D in adjacent sections are automatically connected and considered as a single entity in 3D using ITK's 3D connected filter[40]. ITK is an opensource software widely used for both 2D and 3D medical image analysis. ITK's 3D connected filter is used to merge binary labels in 3D based on overlap of the labels in 3D (adjacent sections post 3D reconstruction). While the Watershed algorithm is often used for merging labels in 3D, it requires multiple parameters (diffusion, gradient, thresholding) to be tuned manually for a dataset that may not generalize in a large dataset like ours. This would result in merging of un-related cells (due to diffusion parameter) or cell dropping (due to gradient threshold parameter). Instead, a connected component filter has been used to merge segmented cells that are locally connected in 2D to create a single unit for each cell. Here, we extend that framework to 3D. No manual tuning of the parameters is required for this connected component filter approach, and ITK's 3D connected filters are routinely used in medical image analysis. All code can be found at GitHub - hubmapconsortium/MATRICS-A: Multiplexed Image Three-D Reconstruction and Integrated Cell Spatial -Analysis, the docker container/environment to run the code can be obtained by using docker pull hubmap/gehc:skin and test data for skin region 7 can be found at Human Digital Twin: Automated Cell Type Distance Computation and 3D Atlas Construction in Multiplexed Skin Biopsies | Zenodo. There is a corresponding ReadMe file that provides context and instructions for the repository's contents and could be found here MATRICS-A/ README.md at main · hubmapconsortium/MATRICS-A · GitHub.

**Statistics and reproducibility**. For patient level comparison, cell counts within the regions of interest (either entire imaged sample or limited to the epidermis region in 3D reconstructed data) were aggregated at patient level for each cell type ($n = 10$ independent datasets for statistical analysis). Then the cell counts were normalized by the volume of the region of interest to account for sample to sample size differences. The number of voxels in the 3D volume were automatically determined using the Insight Toolkit (ITK) software[41]. Using the normalized cell counts aggregated at donor level, statistical hypothesis tests were performed to understand the correlation between the normalized cell count vs. age or UV exposure. To measure the correlation with age, Spearman's correlation was quantified and tested (two-sided). For UV exposure (mild vs. moderate-marked), two-sided Wilcoxon-test was performed. For both statistical tests, Benjamini & Hochberg's multiple testing correction was applied[71]. Statistical analysis was conducted using R version 4.1.2, and additional packages reshape (0.8.9), ggplot2 (3.3.6), and plyr (1.8.7) were used for data processing and visualization.

Violin plots were also included for visualization of cell distance data (i.e., distance of immune cells from nearest endothelial cells (Fig. 5c) and distance of UV damage markers from skin surface (Fig. 6a) These combine a box plot and a density plot to display the probability density of the data at different values. The interquartile range is represented by the box in the center, and the extended line above/below the box shows the upper (max) and lower (min) adjacent values. The median value of the distance is represented by the line in the middle of the box. In the online interactive version, a kernel density estimation is shown on each side of the box to show the distribution shape of the distances of macrophages (CD68+), T helper cells (CD3+ CD4+), T reg cells (CD3+ CD4+ FOXP3+) and T killer cells (CD3+ CD8+) to nearest endothelial cell for each donor https:// hubmapconsortium.github.io/vccf-visualization-release/html/violin_cell.html. The sum of all immune cells grouped by sun exposure is also shown: https:// hubmapconsortium.github.io/vccf-visualization-release/html/violin_cell_all_ region.html.

Similar plots were generated for distance Ki67, p53, and DDB2 positive cells to the skin surface for two regions that were stained using two different cytokeratin cocktails: AE1+ cells (more localized to the stratum basale of the epidermis) and PCK26+ cells (which stained most of entire epidermis). Both cytokeratin cocktails stained cells associated with glandular structures and hair follicles in the dermis.

Distance plots for p53+, Ki67+, and DDB2+ cells in the AE1+/stratum basale epidermis region are located in:
https://hubmapconsortium.github.io/vccf-visualization-release/html/epidermis/ violin_damage_all_region_epidermis.html and in
https://hubmapconsortium.github.io/vccf-visualization-release/html/epidermis/ violin_damage_epidermis.html.

Distance plots for p53+, Ki67+, and DDB2+ cells in the PCK26+/entire epidermis region are located in:
https://hubmapconsortium.github.io/vccf-visualization-release/html/epidermis_ entire/violin_damage_all_region_epidermis.html and in:
https://hubmapconsortium.github.io/vccf-visualization-release/html/epidermis_ entire/violin_damage_epidermis.html

**Cell distance metrics**. Two types of distance analyses were conducted: (1) the distance between the centroid of the immune cell nuclei (macrophage, T helper, T killer, and T reg) and the edge of the nearest blood vessel (endothelial cell) and (2) the distance between the centroid of cells positive for UV damage and repair markers (p53 and DDB2) and proliferation (Ki67) and the nearest edge of the skin surface. To speed up distance calculations for the 13,489 immune cells and 12,407 damage/proliferation markers across the ten 3D regions, a filter was applied to calculate the square root of the distance for only those cells/markers that fell within

the range of the current minimum value of the distance (i.e., cells/markers whose distance on either the x, y, or z axis is greater than the current minimum distance are not included in the distance computation). This approach considerably reduced run time and memory load.

**Interactive visualization of cell and marker distances**. Interactive 3D visualizations of cell and biomarker distances were implemented using the Plotly 3D visualization package, see Fig. 5c. For Ki67, p53, and DDB2 markers within the epidermis and distance to skin surface: https://hubmapconsortium.github.io/vccf-visualization-release/html/epidermis/; for immune cell distances to blood vessels in the dermis: https://hubmapconsortium.github.io/vccf-visualization-release/. The distance calculation result was visualized in two ways: (1) a 3D view projecting all immune cell nuclei, damage/proliferation markers, blood vessels, and their shortest distances (lines) in tissue space; and (2) the histograms showing the distribution of distances between nuclei and blood vessels, and between damage/proliferation markers and the skin surface. The visualization of distance links has been optimized by adding invisible links that unite all of the existing links into a single polyline reducing the size of the vector data and memory usage, allowing for responsive online interaction with about 20,000 nodes in a web browser. In the 3D view, each cell nucleus is rendered as a small circle in the 3D visualization, whereas each UV damage/proliferation marker is rendered as a cross (see legend for cell and biomarker type colors). Endothelial cells (based on CD31 staining) are rendered as a collection of red circles. The outermost skin surface for each tissue section is rendered in grey. The slider in the top-right corner of each 3D view allows the user to view each tissue layer separately. The histogram provides information about the distribution of distances for further analysis. The short lines beneath the histograms indicate the relationship between all the samples and the histogram bars. The histogram can be displayed in three different layouts: Overlaid (by default), Stacked, or Grouped, see selection button on lower left. The visualizations can be exported as an HTML file for online presentation and exploration, or as a vector image for static viewing.

**Immune cell cluster density visualization**. This 3D visualization illustrates the distribution of immune cell clusters, see interactive version at https:// hubmapconsortium.github.io/vccf-visualization-release/html/immune_cluster/. It shows the number of immune cells present within an adjustable radius area, here we show both 15 30 µm. The immune cell cluster view allows for the identification of areas with high immune cell density through use of heatmap-like bubbles of varying sizes and colors. Large, yellow bubbles indicate a high density of immune cells, while small, dark blue bubbles indicate a lower density. The color bar on the right side of the view serves as a legend to assist the reader in identifying the density of the distribution. The slider in the top-right corner of the visualization allows the user to view the cluster view for each tissue layer individually and to compare the 3D and 2D views for further analysis.

**Reporting summary**. Further information on research design is available in the Nature Portfolio Reporting Summary linked to this article.

## Data availability

The anonymized data that support the findings of this study are included in Zenodo https://zenodo.org/record/7565670#.ZDbF_ObMIuV (Original single-cell data for interactive plots—source data for Fig. 4C–E; Fig. 5C, D; Fig. 6A–C. Supplementary Fig. 8A–C) and as a downloadable file from this paper—Supplementary Data 1—source data for Fig. 5A; Supplementary Figs. 6, 7, and 9). All original images for each donor/ region/sequential section are available via publicly accessible HuBMAP Globus sites for each donor/region https://hubmapconsortium.github.io/vccf-visualization-2022/.

## Code availability

All MATRICS-A code can be found at GitHub - hubmapconsortium/MATRICS-A: Multiplexed Image Three-D Reconstruction and Integrated Cell Spatial -Analysis (https://github.com/hubmapconsortium/MATRICS-A). There is a corresponding ReadMe file that provides context and instructions for the repository's contents and can be found at MATRICS-A/README.md at main · hubmapconsortium/MATRICS-A · GitHub. The docker container/environment to run the code can obtained by using docker pull hubmap/gehc:skin. and test data for skin region 7 can be found at https:// zenodo.org/record/7565670#.ZDbF_ObMIuV. Multiple opensource software were used to create the docker container environment and build the code including ITK (ver 5.1), Tensorflow (ver 1.15), Keras (ver 2.2.4), opencv-python (ver 3.4), tifffile (ver 2020.9), Nifty-Reg (ver 1.3), Python (ver 3.6), CMake (ver 3.17). We highly recommend using the docker container to run the code. All VCCF code is available at https://github.com/ hubmapconsortium/vccf-visualization-2022.

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

## Acknowledgements
This work was supported by NIH Award No. 5UH3CA246594-03 (F.G.), OT2OD026671 and OT2OD033756 (K.B.)

## Author contributions
F.G., K.B., S.G., Y.J., J.H., A.S., Y.A.-K., L.D.F., and A.K. conceived the study. E.M.D, C.C., J.M., E.W., and C.S. performed the experimental work related to antibody validation, multiplexing, and data QC. A.C., E.M.D., and C.S. were responsible for the imaging workflows and image data quality and curation. R.R. conducted micro CT analysis of the tissue samples. J.H., A.K., and L.D.F. were responsible for sample collection and histological assessment of the samples. S.G. was responsible for image classification and 3D reconstruction workflows. Y.J. and K.B. were responsible for the spatial cell analysis and development of the interactive plots. S.C. conducted statistical analysis of the data. F.G. and K.B. wrote the first draft of the manuscript. All authors participated in data analysis, manuscript preparation, and editing.

## Competing interests
The authors declare the following competing interests: L.D.F. is founder and has equity interest in SkinJect and Panther Life Sciences. The remaining authors declare no competing interests.
