## [Peer Review File · Communications Biology]

Reviewers' comments:

Reviewer #1 (Remarks to the Author):

Ginty and colleagues presented a 3D Atlas of skins using multiplexed imaging. The authors processed 10 specimens with 26 sequential sections each, and obtained 3D data with 3D reconstruction and single-cell segmentation. From these data, the authors suggested an inverse correlation between DDB2+ cells and age of donors, as well as a positive correlation between T-help/T-killer with age. As great potentials of this study, some serious faults prohibit readers to understand and validate these results. The specific comments are as follows.

Major comments:

(1). The manuscript was put together poorly with various missing components. The figure legends are either simply embedded (Figure 3&5), or completely omitted (Figure 1,2&4, all sup. figures). Without these key information, how could reviewers comprehend the results/figures provided?

(2). Methods are not fully described, or missing key information. For example, while authors mentioned the segmentation were done with "unsupervised GMM", the ref 5 cited in Methods doesn't mention any relevant information. Also, the number of GMM components for each markers isn't well-defined, is that always two (positive or negative)?

(3). The codes/scripts the authors provided are not well-annotated. Different methods (Cell segmentation, classification, 3D reconstruction) are all referred to a single repository, while no further documentation in there to indicate codes/scripts used by different methods.

(4). Though the 3D atlas here could have huge potentials, the shallow sectioning (~130 um) only provides limited view of the whole spacemen. Furthermore, all spacemen are relative small ($14 \times 12 \text{ mm}^2 \sim 47 \times 21 \text{ mm}^2$), the issue with sampling effect might not be neglected. Finally, lacks of 3D or spatial analysis here makes the use of 3D data obsolete. The main conclusion of this study could be derived without 3D or even spatial information, (e.g. flow cytometry of aggregated protein measurements).

(5). Following the point (4), the potential overlapped between adjacent sections wasn't addressed in this manuscript. How does the segmentation or cell classification avoid counting these overlapped cells. How do these methods fuse information from adjacent or nearby sections. It's also unclear that if the segmentation and cell classification was done with 3D or 2D data.

Minor comments

(1). Some figures are incomplete or cut-off (Figure 4A & Figure S4)

(2). Where is the information of 12 cell types used?

(3). The skin-companion website (ref 18) isn't accessible unless the login/credential provided.

(4). In line 442: "The 18-marker panel provided coverage for 9 cell types: epithelial, fibroblast, immune cells (macrophage, T helper, T killer, T reg), nerve, myoepithelial, and endothelial cells." Why not 12?

(5). It would be interesting to see the breakdown DDB2+/Ki67+ in each cell types (Figure 4/S4).

Reviewer #2 (Remarks to the Author):

The manuscript by Ghose, Ju, et al., outlines a workflow for 3D reconstruction and spatial analysis of serial tissue sections labelled using multiplex immunohistochemistry. The study examines cell phenotyping markers in skin biopsy samples to demonstrate this workflow. Multiplex labelling is a powerful tool for investigating tissue structure and changes in disease, and the workflow proposed here is a useful addition to the field and will be of interest to a general readership. The interactive visualisations are particularly engaging and allow for exploration of this high-content spatial data.

I have several considerations that I believe the authors should address.

1. Multiplex labelling relies heavily on appropriate controls to provide confidence in the results. The authors state that "All antibodies used in this study were subjected to a standardized characterization process using a tissue microarray (TMA) and appropriate controls to evaluate the specificity and sensitivity of the primary antibody and its dye-conjugated derivative, including the cyclic testing of the dye inactivation treatment compared to single staining". The reference provided does not outline the TMA validation process. The authors should provide additional description/data/images of their antibody characterisation and controls.

2. It is not clear from the results or methods sections how the microCT images can be used to assist the 3D volume reconstruction. Please expand on this.

3. Please provide more detail on the process for illumination correction and autofluorescence subtraction.

4. Did autofluorescence vary between sections of a biopsy or between biopsy samples? how does this affect the mask for registration? In some tissues, autofluorescence is very high and can colocalise with true label signal, meaning subtraction in post-image processing can delete true signal. Autofluorescence quenching reagents are often used to mitigate this issue during the labelling process, however removing autofluorescence in this way would affect the 3D registration method shown in this manuscript. Therefore, it appears that using an autofluorescence mask for registration could come at the price of sacrificing true labelling signal in tissues with higher autofluorescence. This is a limitation of the method that should be discussed in the context of applying the workflow to other tissues.

5. This study examines different regions of the skin from different individuals. The authors state in the discussion that anatomical location "significantly influences the thickness of the keratin layer, epidermis and dermis, as well as the distribution and density of adnexal structures such as hair follicles, sebaceous, apocrine and sweat glands" and that the data was normalised to total volume, epidermis volume and endothelial cell count to account for these differences. However, these inherent structural differences could affect the response to sun exposure and therefore the results presented in this study. An ideal study examining the difference between samples from mild or moderate sun exposure in young and old individuals would have compared similar biopsy sites. While the primary purpose of this manuscript is obviously to demonstrate the application of the workflow, these limitations in the study design should be acknowledged to aid the readers interpretation of the data.

6. The discussion should include more acknowledgement of the limitations of a serial section reconstruction approach, including issues with tissue deformation, mounting of paraffin sections onto the slide and tissue destruction or loss of a section during cyclic labelling processes. How are these issues managed in the workflow?

Figures

1. Figure 5 – more description is needed in the caption and the text in the images is very small. The interactive visualisations aid the interpretation of this figure immensely and the link should perhaps be included in the caption.

2. Supplementary figures S3, S6 and S7 need more descriptive captions to explain the panels.

3. Supplementary figure S4 is hard to read due to low resolution of the image.

Reviewer #3 (Remarks to the Author):

The authors of the study present MATRICS-A a novel workflow for the reconstruction of 3D tissue from 2D multiplexing layers and calculation of distances from cells to structures of interest. They

applied this framework to a dataset of skin sections with samples from various ages and sun damage levels.

General comment

The main concern of this reviewer is the very limited spatial analyses that were performed potentially missing out on a lot of interesting connections in their data. Most of the results they highlight (cytometric differences) could have been achieved with regular single-cell dissociation-based technologies. The results do not follow the need of generating spatial 3D maps of human tissues (as mentioned in the introduction). The authors of this manuscript presented a very valid and useful framework to reconstruct 3D tissues but this is not sufficiently exploited in the downstream analyses. The manuscript would gain significant impact if this need would be backed up by their results.

Major comments:

Abstract.

The results highlighted in the abstract can be obtained with single-cell dissociation techniques (eg, scRNAseq, cytoF). It is not clear to this reviewer how these results are connected to the need of having 3D spatial data. The authors highlight the ability of the framework to calculate cell distances but then the results are limited to cytometric comparisons. The abstract would read more attractive if the results would highlight the advantage of including 3D spatial data.

Minor:

Abstract

Line 58: Please denote the number of samples.

Introduction

Lines 121-133: Figure 1 legend is integrated into the text which reads confusing.

Results

Lines 150-152: Please add the rationale behind the selection of cell-type/marker subset from the whole table is missing. Are these the more abundant cell types or is there any other reason to select specifically those?

Lines 159: The authors claim that it would have been cumbersome and time-consuming to design this study with only deep learning models. However, there are already pre-trained models that with little fine tuning provide reasonably good results (see for example StarDist3D: https://openaccess.thecvf.com/content_WACV_2020/html/Weigert_Star-convex_Polyhedra_for_3D_Object_Detection_and_Segmentation_in_Microscopy_WACV_2020_paper.html). There are also frameworks to prioritise the images that perform poorly and require better annotations (https://onlinelibrary.wiley.com/doi/10.1002/cjp2.229?utm_content=buffer3a40d&utm_medium=social&utm_source=linkedin.com&utm_campaign=buffer). This should be at least introduced in the discussion.

Lines 188-191: The authors claim that their registration approach improves accuracy in a number of scenarios. The claim should be supported by results or literature.

GE
Research

Fiona Ginty PhD
Senior Principal Scientist
Biology & Applied Physics

Detailed Response to the Editor and Reviewers:

Editor Summary:

While the referees find your work of some interest, they raise concerns about the strength of the novel conclusions that can be drawn at this stage.

We appreciate that the issues raised regarding missing information and clarity in the presentation of the workflow could potentially be addressed in a revision. However, we feel that the comments from both reviewer 1 and 3 that the conclusions from the skin-sections could have been drawn by using more conventional methods prevent us from pursuing the paper further. As you can see the reviewers also raise lack of certain controls and validation.

We feel that these reservations are sufficiently important as to preclude publication of this study in Communications Biology.

Author Response:

Thank you for the opportunity to submit a substantially revised paper. The paper now provides quantitative evidence for the value of 3D data over 2D data. Specifically:

- *We demonstrate that the average distance between immune cells and endothelial cells is roughly half in 3D vs 2D (~56 μm vs 108 μm).*
- *We quantified 10-70% more T cells within 30 μm of a T helper cell in 3D vs 2D. These differences are an important consideration for samples with low immune cell density, as analysis of spatial relationships and cell-cell interactions in 2D would be more challenging to accurately quantify.*
- *The novel application of immune cell cluster density plots (inspired by 3D geospatial and population density plots) clearly illustrates the variations in immune cell density within and across samples.*
- *Distance of DNA damage and proliferation markers to the skin surface (shorter distance is a measure of cancer risk) did not differ by age or sun exposure. These markers were largely localized the stratum basale (the lower layer of the epidermis) and in three cases were found deeper in the dermis and associated with hair follicle regions (illustrated by new spatial distance plots). A shorter distance to the skin surface would indicate higher risk of skin cancer.*
- *Our integrated cell segmentation and 3D reconstruction workflow is unique in that it merges cells that partially appear in adjacent sections, which is critical for accurate spatial cell analysis in 3D.*
- *MATRICES-A software provides an end-to-end solution for segmentation and classification and is available open access, now with updated readme instructions and test data for evaluation.*
- *Our updated companion website provides complete details of the data and visualization tools Companion Website for "Human Digital Twin: 3D Atlas Reconstruction of Skin and Spatial Mapping of Immune Cell Density, Vascular Distance and Effects of Sun Exposure and Aging".*
- *We now provide details on the antibody validation process and address all other issues raised by the reviewers (please see below).*

Reviewer #1 (Remarks to the Author):

Ginty and colleagues presented a 3D Atlas of skins using multiplexed imaging. The authors processed 10 specimens with 26 sequential sections each and obtained 3D data with 3D reconstruction and single-cell segmentation. From these data, the authors suggested an inverse correlation between DDB2+ cells and age of donors, as well as a positive correlation between T-help/T-killer with age. As great potentials of this study, some serious faults prohibit readers to understand and validate these results. The specific comments are as follows.

- **Author response:** *We appreciate your thoughtful review and insightful comments. We have made a concerted effort to address all your concerns.*

Major comments:

1. The manuscript was put together poorly with various missing components. The figure legends are either simply embedded (Figure 3&5), or completely omitted (Figure 1,2&4, all sup. figures). Without these key information, how could reviewers comprehend the results/figures provided?

- **Response:** *We apologize for these omissions. The substantially revised paper now describes the key results (not embedded) in text and legends for all main figures, supplementary figures, and tables.*

2. Methods are not fully described or missing key information.

- **Response:** *We added details to the Methods section as follows:*

- **Antibody validation:** *Details and links to protocols in protocols.io are now provided for the antibody validation process in the **Methods (lines 530-560)**. We have also included two images for each antibody from a reference multi-organ tissue array (MTU391, Pantomics), which has been consistently used to characterize all antibodies (>500) over the last decade and a corresponding example image from skin using the same region of interest for comparison (**Supp. Fig. S11 A-R**).*
- **Multiplexed staining and imaging:** *Additional details and links to the multiplexed staining protocols in protocols.io are now provided in the **Methods (lines 562-575)**.*
- **Illumination Correction:** *We now provide a description of the illumination correction procedure (also requested by Reviewer 3) in the **Methods (lines 577-594)**. Two citations were provided for autofluorescence subtraction (previously described in supporting information of Gerdes MJ et al. Highly multiplexed single-cell analysis of formalin-fixed, paraffin-embedded cancer tissue. Proc Natl Acad Sci. 2013 Jul 16;110(29):11982-7. doi: 10.1073/pnas.1300136110 and in Woolfe F, Gerdes M, Bello M, Tao X, Can A. Autofluorescence removal by non-negative matrix factorization. IEEE Trans Image Process. 2011 Apr;20(4):1085-93. doi: 10.1109/TIP.2010.2079810).*
- **Image Normalization:** *A description of image normalization prior to deep learning-based nuclei segmentation and biomarker probability estimation using GMM is now included in the **Methods (lines 596-604)**.*
- **Annotations for quantifying cell classification accuracy:** *Now separated out into a separate section in **Methods (lines 644-648)**.*

- **3D reconstruction and merging overlapping cells:** the overall description in the **Results (lines 236-241)** and **Methods (lines 672-682)** has been refined to ensure clarity and completeness and details on the method for merging overlapping cells is provided in each section.
 - **Visualization of immune cell clusters in 3D:** a new method has been applied that allows visualization of immune cell cluster density in 3D (**Methods, lines 754-773**).
3. For example, while authors mentioned the segmentation were done with "unsupervised GMM", the ref 5 cited in Methods doesn't mention any relevant information. Also, the number of GMM components for each markers isn't well-defined, is that always two (positive or negative)?
- **Response:** The correct citation for GMM approach has now been added to line 199 (Reynolds, D. Gaussian Mixture Models. in Encyclopedia of Biometrics (Springer, 2009). Details of GMM component definition are now provided in **Methods (line 621-625)**. Specifically, GMM with two clusters was used to obtain a probabilistic segmentation of the cell-type and functional markers and background. Pixels with high signal for cell type and functional markers were automatically assigned high probability values in one cluster (positive class) and pixels with low signal intensity were assigned high probability values in the second cluster (background – negative class).
4. The codes/scripts the authors provided are not well-annotated. Different methods (Cell segmentation, classification, 3D reconstruction) are all referred to a single repository, while no further documentation in there to indicate codes/scripts used by different methods.
- **Response:** We have substantially improved software documentation of MATRICS-A. All code can be found at <https://github.com/hubmapconsortium/MATRICES-A>, the docker container/environment to run the code can be obtained by using `docker pull hubmap/gehc:skin` and test data for skin region 7 can be found at <https://zenodo.org/record/7565670#.Y9EoSS-B2-p>. There is a corresponding **ReadMe file** that provides context and instructions for the repository's contents at: <https://github.com/hubmapconsortium/MATRICES-A/blob/main/README.md>. Directions are now clearly described in the **Methods** section under Cell Segmentation and Classification (**lines 634-640**) and 3D Reconstruction (**lines 682-688**).
5. Though the 3D atlas here could have huge potentials, the shallow sectioning (~130 um) only provides limited view of the whole specimen. Furthermore, all specimens are relatively small (14x12 mm² ~ 47x21 mm²), the issue with sampling effect might not be neglected. Finally, lacks of 3D or spatial analysis here makes the use of 3D data obsolete. The main conclusion of this study could be derived without 3D or even spatial information, (e.g. flow cytometry of aggregated protein measurements).
- **Response:** Thank you for raising these concerns. While we focused on 26 serial sections, the MATRICS-A image registration pipeline is not limited to a sectioning thickness or quantity of sections. We can also theoretically use the same pipeline for cell segmentation and 3D reconstruction for an expanded number of sections (technically there is no limit, other than cost and time).

- Regarding the value of 3D over 2D analysis: Given your expert comments, we run additional analyses and implemented novel data visualizations which clearly show the value of using 3D over 2D. This type of analysis could not be done with flow cytometry since it would not provide spatial distribution of cells within the tissue and in relation to neighboring cells. Specifically:
 - We demonstrate that the average distance between immune cells and endothelial cells is roughly half in 3D vs 2D (~56 μm vs 108 μm).
 - We quantified 10-70% more T cells within 30 μm of a T helper cell in 3D vs 2D. These differences are an important consideration for samples with low immune cell density, as analysis of spatial relationships and cell-cell interactions in 2D would be more challenging to accurately quantify.
 - The novel application of immune cell cluster density plots (inspired by 3D geospatial and population density plots) clearly illustrates the variations in immune cell density within and across samples.
 - Distance of DNA damage and proliferation markers to the skin surface (shorter distance is measure of cancer risk) did not differ by age or sun exposure. These markers were largely localized the stratum basale (the lower layer of the epidermis) and in three cases were found deeper in the dermis and associated with hair follicle regions (illustrated by new spatial distance plots. A shorter distance to the skin surface would indicate higher risk of skin cancer.
- Specific edits are as follows:
 - **Lines 291-303: T cell density is lower in 2D vs 3D volumes:** We then compared immune cell density in 2D vs 3D as 1) the average number of T cells within 30 μm of a T helper cell (**Figure 4A**) and 2) the maximum number of T cells within 30 μm of a T helper cell (**Figure 4B**). There was wide variation in both measurements across all samples due to heterogeneous cell density in each section/sample. For example, regions 1 and 2 had similar maximum number of immune cells ($n=3$) within 30 μm of a T helper cell in 2D and 3D, region 7 had a maximum of 11 cells in 3D, and just 3 cells in 2D. Overall, we quantified 10-70% more T cells within 30 μm of a T helper cell in 3D vs 2D. This difference is important for samples with low immune cell density, where analysis of spatial relationships and cell-cell interactions in 2D would be more challenging to accurately quantify. The immune cell cluster density plots in **Figs. 4C-E** illustrate three contrasting examples of skin regions with low (region 1), medium-high (region 9) and high (region 12) immune cell cluster density in 3D, respectively. The low and high examples may be attributed to the health/therapy status of the donors who were noted as having rheumatoid arthritis (region 1) and systemic lupus erythematosus (region 12).
 - **Lines 310-319: Shorter distances between immune cells and endothelial cells in 3D vs 2D.** There were significant differences between 2D and 3D in the average distance of the nearest endothelial cell to immune cells (macrophages, T helper, T killer and T regs), with distances in 3D typically much shorter than 2D (~56 μm vs 108 μm on average, ($p<0.0001$) (**Fig. 5A**). Distances (in 3D) between each immune cell type and endothelial cells are also shown as violin plots and grouped by age and sun exposure for each donor/region in **Fig. 5B**. There was a trend for higher counts of T killer cells within 100 μm of endothelial cells in younger donors (corr=-0.73, adjusted p -value=0.08; see **Supp. Fig S7iii**). The implications of this are unclear without further validation in a larger group of subjects but may reflect age-related differences in adaptive immune response. An example of a region with higher total immune cell counts (including T killer) within 100 μm of endothelial cells is shown in **Fig. 5C**.

- **Lines 321-342: No differences in spatial location of sun damage/proliferation cell markers age or sun exposure.** We quantified distance of p53, Ki67 and DDB2 positive keratinocytes to the skin surface using two different epidermis masks: 1) using AE1 cytokeratin cocktail, which was more specific for the lower epidermis/stratum basale layer and hair follicular units; and 2) CK26 cocktail, which stained the entire epidermis, as well as hair follicular units (example images comparing the staining characteristics are shown in **Supp. Fig. S11B**). Due to non-uniformity of the skin surface, the spatial cell distance analysis was conducted using a hybrid of 3D and 2D data, whereby the distances of the 3D reconstructed cells to the skin surface were calculated using the nearest 2D tissue section. We found that most Ki67 and p53 positive keratinocytes were largely localized to the AE1+/stratum basale region (where regenerating keratinocyte stem cells are localized⁴⁸). There were no significant differences in distance of p53, DDB2 and Ki67 positive keratinocytes to the skin surface when analyzed by sun exposure or aging. Notably, there were three cases (regions 1, 2 and 9) that had a very wide spatial distribution of p53, Ki67 positive cells (up to 1600 μm from the skin surface (**Supp. Fig. S8A-C**). In each case, this was due a hair follicular unit extending deeper in the dermis with a high number of p53 and Ki67 positive cells. The number of p53 positive cells has been shown to extend deeper into the hair follicles and glands in older patients⁴⁹ (these samples were from donors aged 52-72 years, however we did not have matched younger patients for comparison). In all other cases, the average distance of p53 and Ki67 cells from the skin surface was 155 μm and 143 μm , respectively. Total Ki67 and p53 positive cell count was not significantly correlated with age or sun exposure (**Supp. Fig. S9**)

6. The potential overlapped between adjacent sections wasn't addressed in this manuscript. How does the segmentation or cell classification avoid counting these overlapped cells.

- **Response:** Excellent point. The revised paper now presents a details in **Methods (lines 671-682)**: Cells overlapping in 3D in adjacent sections are automatically connected and considered as a single entity in 3D using ITK's 3D connected filter³⁸. ITK is an open source software widely used for both 2D and 3D medical image analysis. ITK's 3D connected filter is used to merge binary labels in 3D based on overlap of the labels in 3D (adjacent sections post 3D reconstruction). While the Watershed algorithm is often used for merging labels in 3D, it requires multiple parameters (diffusion, gradient, thresholding) to be tuned manually for a dataset that may not generalize in a large dataset like ours. This would result in merging of un-related cells (due to diffusion parameter) or cell dropping (due to gradient threshold parameter). Instead, a connected component filter has been used to merge segmented cells that are locally connected in 2D to create a single unit for each cell. Here, we extend that framework to 3D. No manual tuning of the parameters is required for this connected component filter approach, and ITK's 3D connected filters are routinely used in medical image analysis.
- Also summarized in **Results - Line 237-241** - We use ITK's 3D connected component image filter to merge overlapping cells and classify cells in 3D^{39,40}. Connected component image filter has historically been used in merging segmentations in 3D⁴¹⁻⁴³ and for refining cell segmentation in 2D⁴⁴. To the best of our knowledge, this is the first time a 3D connected component has been used to fuse overlapping cells in 3D in serial histological sections.

7. How do these methods fuse information from adjacent or nearby sections.

- **Response:** Thank you for asking this critical question. As explained in #6 above, we use a 3D connected component filter to fuse overlapping cells in 3D. Cells are first segmented and classified into different cell types 2D. The 2D segmented cells are then reconstructed in 3D using the transformation map generated in the registration framework. Cells overlapping in 3D in adjacent sections are automatically connected and considered as a single entity in 3D using ITK's 3D connected filter (as described above and updated in **Methods (lines 671-682)**).
8. It's also unclear that if the segmentation and cell classification was done with 3D or 2D data.
- **Response:** Thank you for your feedback. We have now added details to further clarify which steps were done in 2D versus 3D. We first segmented and classified cells in 2D and compare the segmentation to manual annotations to quantify sensitivity and specificity and for QC. We then proceeded to 3D reconstruction of the 2D segmented cells using the 3D registration framework and further refined cell segmentation in 3D using 3D connected component filter (ITK). Please refer to **Line 194-196** for clarification: Our method provides an integrated workflow for 2D segmentation and classification of cell types from the multiplexed images, followed by automated 3D reconstruction.

Minor comments

1. Some figures are incomplete or cut-off (Figure 4A & Figure S4)
 - **Response:** All figures have been checked for quality and updated and detailed legends have been included throughout. **Figure 4A (now Figure 6A)** has been expanded to accommodate the full range of data. The average distance of p53 and Ki67 positive cells was 155 μm and 143 μm , from the skin surface, however there was a wider distance for 3 donors who also had positive cells in hair follicles, which were located up to 1600 μm from the skin surface, in the dermis regions. A link to further interactive visualization is available at https://hubmapconsortium.github.io/vccf-visualization-release/html/epidermis_entire/violin_damage_epidermis.html
 - Figure S4 was intended to be a limited snapshot of the ASCT+B skin report as the full report would require multiple letter size pages. A link to the skin report is now provided so that readers can go directly there to review the entire report: ASCT+B Skin Report.
2. Where is the information of 12 cell types used?
 - **Response:** We have now included a table (**Supp. Table S2**) which provides a detailed breakdown of markers and cell types. After consultation with a dermatologist expert and literature, we increased the number of cell types to 14 because the pan cytokeratin cocktails that we used (CK26 (KRT1, KRT5, KRT6, KRT8) and AE1 (KRT10, KRT14, KRT15, KRT16 and KRT19) have broader cell coverage. However, additional markers will be added in future work to increase specificity (e.g., keratin 10 for granular keratinocytes). Additional details have now been provided in the results: **Construction of a Skin Anatomical Structure and Cell Type + Biomarker (ASCT+B) Table**, line 176-184) and **Methods (Antibody validation, lines 530-560)** and a new table has been added that more clearly shows the alignment between the biomarkers and cell types that were included in this study (**Supplementary Table S2**).

3. The skin-companion website (ref 18) isn't accessible unless the login/credential provided.
 - **Response:** *Unfortunately, this reference was for the google doc version which was password protected. We have now updated the link to the published/public facing version: Companion Website for "Human Digital Twin: 3D Atlas Reconstruction of Skin and Spatial Mapping of Immune Cell Density, Vascular Distance and Effects of Sun Exposure and Aging"*
4. In line 442: "The 18-marker panel provided coverage for 9 cell types: epithelial, fibroblast, immune cells (macrophage, T helper, T killer, T reg), nerve, myoepithelial, and endothelial cells." Why not 12?
 - **Response:** *This has now been corrected. As explained above, after consultation with a Dermatologist expert and literature, we increased the number of cell types to 14 because the pan cytokeratin cocktails that we used (CK26 (KRT1, KRT5, KRT6, KRT8) and AE1 (KRT10, KRT14, KRT15, KRT16 and KRT19) have broader cell coverage. However, additional markers will be added in future work to increase specificity (e.g., keratin 10 for granular keratinocytes). Additional details have now been provided in the results: **Construction of a Skin Anatomical Structure and Cell Type + Biomarker (ASCT+B) Table**, line 176-184) and **Methods (Antibody validation**, lines 530-560) and a new table has been added that more clearly shows the alignment between the biomarkers and cell types that were included in this study (**Supplementary Table S2**).*
5. It would be interesting to see the breakdown DDB2+/Ki67+ in each cell types (Figure 4/S4).
 - **Response:** *Thank you very much for the suggestion. We agree that this would be very interesting, and we plan to incorporate into our follow-up analysis with a larger number of samples.*

Reviewer #2 (Remarks to the Author):

The manuscript by Ghose, Ju, et al., outlines a workflow for 3D reconstruction and spatial analysis of serial tissue sections labelled using multiplex immunohistochemistry. The study examines cell phenotyping markers in skin biopsy samples to demonstrate this workflow. Multiplex labelling is a powerful tool for investigating tissue structure and changes in disease, and the workflow proposed here is a useful addition to the field and will be of interest to a general readership. The interactive visualisations are particularly engaging and allow for exploration of this high-content spatial data. I have several considerations that I believe the authors should address.

1. Multiplex labelling relies heavily on appropriate controls to provide confidence in the results. The authors state that "All antibodies used in this study were subjected to a standardized characterization process using a tissue microarray (TMA) and appropriate controls to evaluate the specificity and sensitivity of the primary antibody and its dye-conjugated derivative, including the cyclic testing of the dye inactivation treatment compared to single staining". The reference provided does not outline the TMA validation process. The authors should provide additional description/data/images of their antibody characterisation and controls.

- **Response:**
 - **Antibody validation:** Thank you for the helpful feedback. Details and links to protocols in protocols.io now provided for the antibody validation process in the **Methods (lines 530-560)**. We have also included two images for each antibody from a reference multi-organ tissue array (MTU391, Pantomics), which has been consistently to characterize all antibodies (>500) over the last decade and a corresponding example image from skin using the same region of interest for comparison (**Supp. Fig. S11 A-R**).

- 2. It is not clear from the results or methods sections how the microCT images can be used to assist the 3D volume reconstruction. Please expand on this.
 - **Response:** Thank you for highlighting this gap. In the absence of an external 3D reference, 2D to 3D reconstruction suffers from alignment problem often referred to as “banana effect” (where there is a straightening of curved structures during registration). Hence, we obtain 3D CT prior to sectioning and use that as an external reference for our 3D reconstruction method. We now describe in the **Introduction** and discuss in more detail in the **Discussion**:
 - **Introduction (Lines 124-132):** In general, reconstruction of 3D volumes from 2D serial sections is a complex procedure and can suffer from the “banana effect” (where curved structures are incorrectly straightened during image registration) in absence of external reference structures^{20,21}. Further, the 3D reconstruction process tends to be computationally slow and requires significant manual intervention to ensure robust alignment, which is essential for accurate 3D spatial cell analysis. To address these challenges, we have developed an automated, reproducible workflow (Multiplexed Image Three-D Reconstruction and Integrated Cell Spatial - Analysis - MATRICS-A) for 3D reconstruction of highly multiplexed tissue sections. Compared to previous 3D reconstruction methods¹⁴⁻¹⁷, our approach is calibrated using micro CT images of the formalin fixed block, thus improving 3D reconstruction accuracy (and reducing the “banana effect”).
 - **Discussion (lines 398-407):** As described earlier, one limitation of 3D reconstruction of tissue sections has been the introduction of the “banana effect” which erroneously straightens curved anatomical structures during registration. Micro CT imaging of the tissue block prior to sectioning, provides a reference volume to which the 3D reconstructed volume of the skin tissue and the cells are registered. A similar approach has also been successfully applied for whole human brains using MRI, blockface photography (to bridge the gap between MRI and histology) and thick sequential histological brain sections (25 μm thick)¹⁷.

- 3. Please provide more detail on the process for illumination correction and autofluorescence subtraction.
 - **Response: Illumination Correction:** We now provide a description of the illumination correction procedure in the **Methods (lines 577-594)**. Two citations were provided for **autofluorescence subtraction** (provided in supporting information of Gerdes MJ et al. Highly multiplexed single-cell analysis of formalin-fixed, paraffin-embedded cancer tissue. *Proc Natl Acad Sci.* 2013 Jul 16;110(29):11982-7. doi: 10.1073/pnas.1300136110 and in Woolfe F, Gerdes M, Bello M, Tao X, Can A. Autofluorescence removal by non-negative matrix factorization. *IEEE Trans Image Process.* 2011 Apr;20(4):1085-93. doi: 10.1109/TIP.2010.2079810).

4. Did autofluorescence vary between sections of a biopsy or between biopsy samples? how does this affect the mask for registration? In some tissues, autofluorescence is very high and can co-localize with true label signal, meaning subtraction in post-image processing can delete true signal. Autofluorescence quenching reagents are often used to mitigate this issue during the labelling process, however removing autofluorescence in this way would affect the 3D registration method shown in this manuscript. Therefore, it appears that using an autofluorescence mask for registration could come at the price of sacrificing true labelling signal in tissues with higher autofluorescence. This is a limitation of the method that should be discussed in the context of applying the workflow to other tissues.
- **Response:** *Thank you for raising this excellent point. We use Otsu thresholding to segregate foreground and background AF signal. Otsu thresholding uses intra class variance computed from image histogram to set the thresholds for the background and the foreground. Hence while the AF signal distribution may change from one serial section to another, the threshold is set based on intra class variance and is automatically adjusted from one serial section to another to maximize the separation of the background and the foreground. Additional details and clarification have now been added to **Methods (lines 650-667)**.*
 - *To address the valid point about quenching methods for AF and colocalization of AF with label signal, none of our samples underwent quenching to reduce signal and all AF images used for reconstruction were unstained (i.e., the image is taken prior to any marker staining), hence AF-based registration is independent of biomarker signal or signal variations associated with staining. This has now been clarified in the results section (lines 229-231). The use of quenching is highlighted in the discussion as a consideration for study design in the **Discussion (lines 388-392)**.*
5. This study examines different regions of the skin from different individuals. The authors state in the discussion that anatomical location “significantly influences the thickness of the keratin layer, epidermis and dermis, as well as the distribution and density of adnexal structures such as hair follicles, sebaceous, apocrine and sweat glands” and that the data was normalised to total volume, epidermis volume and endothelial cell count to account for these differences. However, these inherent structural differences could affect the response to sun exposure and therefore the results presented in this study. An ideal study examining the difference between samples from mild or moderate sun exposure in young and old individuals would have compared similar biopsy sites. While the primary purpose of this manuscript is obviously to demonstrate the application of the workflow, these limitations in the study design should be acknowledged to aid the readers interpretation of the data.
- **Response:** *We agree with the reviewer and have more clearly highlighted this limitation in the **Discussion (lines 425-431)**: One important consideration when interpreting volumetric immune cell counts is anatomical location, which significantly influences the thickness of keratin layer, epidermis and dermis, as well as the distribution and density of adnexal structures such as hair follicles, which are significantly exposed to the sun when compared to sites such as trunk, medial aspect of extremities (inner thigh, arm, etc.) and plantar foot (sole), leading to marked differences in the amount of cumulative UV exposure and skin damage. Partially addressing this, our samples were collected from across the anatomy, including arms, legs, abdomen, and scalp and normalized for volume and endothelial cell count to account for sample-to-sample differences. However, an ideal study design*

would be to prospectively collect at least two distinct anatomical samples from the same donor and expand racial diversity, which is planned as part of future efforts on HuBMAP.

6. The discussion should include more acknowledgement of the limitations of a serial section reconstruction approach, including issues with tissue deformation, mounting of paraffin sections onto the slide and tissue destruction or loss of a section during cyclic labelling processes. How are these issues managed in the workflow?

- **Response:** We appreciate the feedback and have included the following guidance and recommendations on experimental design considerations. **Discussion lines 373-396:** There are several experimental design factors to consider when planning 3D reconstruction of serial sections using multiplexed immunofluorescence images. For greatest efficiency (cost/time), we embedded 12 samples in a single paraffin block, which also allowed consistent staining and imaging across all samples. Although there is some trade off in terms of reduced spatial cell heterogeneity, in this instance, this was offset by having a diverse range of samples from a wide age range and sun exposure effects and anatomical location. The blocks were cut into 100 x 5 μm sections and 26 of the highest quality sections were down selected based on correct placement on the slide and no visual imperfections such as tears or missing tissue. This is important for multiplexed imaging processes where the coverslip is removed between each staining round and may result in damage to the tissue (in our experience, if tissue loss occurs, it typically takes place in the first 1-3 rounds of staining, and especially if samples have small tears due to fragile nature or sectioning). Use of Superfrost™ slides is an important mitigation against tissue loss. More recent commercial options for multiplexing include the use of a flow cell or coverslip-free format, hence tissue loss may be less of a concern for future work. Serial sections were all stained in a single batch with well characterized antibodies, providing high quality, consistent images. The AF images that were used for 3D reconstruction were all taken prior to biomarker staining, providing a “clean” AF image with no colocalization of stained biomarker signal. Quenching methods are sometimes used to reduce signal in high AF tissues (e.g. Sudan black B treatment has been reported to reduce AF signal by 60-95% in pancreatic tissue⁵²), however we did not use quenching (background is imaged separately and subtracted from true biomarker signal) and further evaluation is needed to determine if quenching would negatively affect this AF-based registration workflow. Although there is inherent variation in AF signal from section to section or from sample to sample, we address this using the Otsu thresholding approach, which uses intra class variance to set the AF threshold for each sample/slide. The AF threshold is automatically adjusted from one serial section to another to maximize the separation of the background and the foreground.

- **To further mitigate against section to section variability, we also address the benefits of using micro CT in providing an external reference for 3D reconstruction: Discussion lines 404-413:** Posterior registration/alignment of the 3D reconstructed volumes to micro CT images is valuable when there is deformation and/or wear and tear in the tissue samples during the cyclic staining process. The co-registration maps microfeatures (e.g., cell types) to macro imaging features, and compared to landmark-constrained 3D histological imaging⁵³, minimal manual intervention is required for accurate registration and reconstruction of the 3D volume. Within our workflow, a single slide from the entire volume is identified manually based on tissue quality as the reference image; the rest of the process for registration and 3D reconstruction is completely automatic. Compared to image similarity-based alignment^{14,40}, automatic block correspondences is used for initial affine alignment. Use of block

correspondences improves registration speed compared to image similarity-based alignment methods^{38,40} and also accounts for local deformations that may happen during the staining and image acquisition process.

7. Figure 5 – more description is needed in the caption and the text in the images is very small. The interactive visualisations aid the interpretation of this figure immensely and the link should perhaps be included in the caption.
 - **Response:** *additional descriptions have been provided for all figures, including the links to all the interactive visualizations.*
8. Supplementary figures S3, S6 and S7 need more descriptive captions to explain the panels.
 - o **Response:** *Additional information has been provided to the captions of these figures (now Supp Figs S4 (previously S3) and S5A and B (previously S6 and S7)).*
9. Supplementary figure S4 is hard to read due to low resolution of the image.
 - o **Response:** *This figure is now Supp. Fig S3 and has been replaced with a higher resolution image and link to the complete ASCT+B reporter for skin - ASCT+B Skin Report*

Reviewer #3 (Remarks to the Author):

The authors of the study present MATRICS-A a novel workflow for the reconstruction of 3D tissue from 2D multiplexing layers and calculation of distances from cells to structures of interest. They applied this framework to a dataset of skin sections with samples from various ages and sun damage levels.

General comment

1. The main concern of this reviewer is the very limited spatial analyses that were performed potentially missing out on a lot of interesting connections in their data. Most of the results they highlight (cytometric differences) could have been achieved with regular single-cell dissociation-based technologies. The results do not follow the need of generating spatial 3D maps of human tissues (as mentioned in the introduction). The authors of this manuscript presented a very valid and useful framework to reconstruct 3D tissues but this is not sufficiently exploited in the downstream analyses. The manuscript would gain significant impact if this need would be backed up by their results.
 - **Response:** *We appreciate this expert feedback. We conducted additional analysis to highlight important differences between 2D and 3D and developed an additional novel cluster density plot to highlight cell density in 3D. This type of analysis could not be done with flow cytometry since that technology platform would not provide spatial distribution of cells within the tissue. The key new findings include the following:*
 - o *We demonstrate that the average distance between immune cells and endothelial cells is roughly half in 3D vs 2D (~56 μ m vs 108 μ m).*

- We quantified 10-70% more T cells within 30 μm of a T helper cell in 3D vs 2D. These differences are an important consideration for samples with low immune cell density, as analysis of spatial relationships and cell-cell interactions in 2D would be more challenging to accurately quantify.
- The novel application of immune cell cluster density plots (inspired by 3D geospatial and population density plots) clearly illustrates the variations in immune cell density within and across samples.
- Distance of DNA damage and proliferation markers to the skin surface (shorter distance is measure of cancer risk) did not differ by age or sun exposure. These markers were largely localized the stratum basale (the lower layer of the epidermis) and in three cases were found deeper in the dermis and associated with hair follicle regions (illustrated by new spatial distance plots (**Supp Fig. 8A-C**). A shorter distance to the skin surface would indicate higher risk of skin cancer.
- Specific line edits are as follows:
 - **Lines 291-303: T cell density is lower in 2D vs 3D volumes:** We then compared immune cell density in 2D vs 3D as 1) the average number of T cells within 30 μm of a T helper cell (**Figure 4A**) and 2) the maximum number of T cells within 30 μm of a T helper cell (**Figure 4B**). There was wide variation in both measurements across all samples due to heterogeneous cell density in each section/sample. For example, regions 1 and 2 had similar maximum number of immune cells ($n=3$) within 30 μm of a T helper cell in 2D and 3D, region 7 had a maximum of 11 cells in 3D, and just 3 cells in 2D. Overall, we quantified 10-70% more T cells within 30 μm of a T helper cell in 3D vs 2D. This difference is important for samples with low immune cell density, where analysis of spatial relationships and cell-cell interactions in 2D would be more challenging to accurately quantify. The immune cell cluster density plots in **Figs. 4C-E** illustrate three contrasting examples of skin regions with low (region 1), medium-high (region 9) and high (region 12) immune cell cluster density in 3D, respectively. The low and high examples may be attributed to the health/therapy status of the donors who were noted as having rheumatoid arthritis (region 1) and systemic lupus erythematosus (region 12).
 - **Lines 310-319: Shorter distances between immune cells and endothelial cells in 3D vs 2D.** There were significant differences between 2D and 3D in the average distance of the nearest endothelial cell to immune cells (macrophages, T helper, T killer and T regs), with distances in 3D typically much shorter than 2D ($\sim 56 \mu\text{m}$ vs $108 \mu\text{m}$ on average, ($p < 0.0001$) (**Fig. 5A**). Distances (in 3D) between each immune cell type and endothelial cells are also shown as violin plots and grouped by age and sun exposure for each donor/region in **Fig. 5B**. There was a trend for higher counts of T killer cells within 100 μm of endothelial cells in younger donors ($\text{corr} = -0.73$, adjusted $p\text{-value} = 0.08$; see **Supp. Fig S7iii**). The implications of this are unclear without further validation in a larger group of subjects but may reflect age-related differences in adaptive immune response. An example of a region with higher total immune cell counts (including T killer) within 100 μm of endothelial cells is shown in **Fig. 5C**.
 - **Lines 321-342: No differences in spatial location of sun damage/proliferation cell markers age or sun exposure.** We quantified distance of p53, Ki67 and DDB2 positive keratinocytes to the skin surface using two different epidermis masks: 1) using AE1 cytokeratin cocktail, which was more specific for the lower epidermis/stratum basal layer

and hair follicular units; and 2) CK26 cocktail, which stained the entire epidermis, as well as hair follicular units (example images comparing the staining characteristics are shown in **Supp. Fig. S11B**). Due to non-uniformity of the skin surface, the spatial cell distance analysis was conducted using a hybrid of 3D and 2D data, whereby the distances of the 3D reconstructed cells to the skin surface were calculated using the nearest 2D tissue section. We found that most Ki67 and p53 positive keratinocytes were largely localized to the AE1+/stratum basale region (where regenerating keratinocyte stem cells are localized⁴⁸). There were no significant differences in distance of p53, DDB2 and Ki67 positive keratinocytes to the skin surface when analyzed by sun exposure or aging. Notably, there were three cases (regions 1, 2 and 9) that had a very wide spatial distribution of p53, Ki67 positive cells (up to 1600 μm from the skin surface (**Supp. Fig. S8A-C**)). In each case, this was due a hair follicular unit extending deeper in the dermis with a high number of p53 and Ki67 positive cells. The number of p53 positive cells has been shown to extend deeper into the hair follicles and glands in older patients⁴⁹ (these samples were from donors aged 52-72 years, however we did not have matched younger patients for comparison). In all other cases, the average distance of p53 and Ki67 cells from the skin surface was 155 μm and 143 μm , respectively. Total Ki67 and p53 positive cell count was not significantly correlated with age or sun exposure (**Supp. Fig. S9**

Major comments:

Abstract.

1. The results highlighted in the abstract can be obtained with single-cell dissociation techniques (eg, scRNAseq, cytoF). It is not clear to this reviewer how these results are connected to the need of having 3D spatial data. The authors highlight the ability of the framework to calculate cell distances but then the results are limited to cytometric comparisons. The abstract would read more attractive if the results would highlight the advantage of including 3D spatial data.

- **Response:** We now have updated the abstract to highlight our updated spatial analysis and differences in 2D and 3D.

Minor:

Abstract

2. Line 58: Please denote the number of samples.

- **Response:** this has now been updated to state 26 serial sections (**now line 60**).

3. Introduction

Lines 121-133: Figure 1 legend is integrated into the text which reads confusing.

- **Response:** This text has now been removed.

Results

4. Lines 150-152: Please add the rationale behind the selection of cell-type/marker subset from the whole table is missing. Are these the more abundant cell types or is there any other reason to select specifically those?

- **Response:** the rationale for marker selection has been updated and was based on both providing coverage for the major cell types and biomarkers that have been documented in the skin ASCT+B tables. It was also directed by the types of analysis we sought to undertake:
 - o **Lines 169-177:** Here we focus on a subset of the skin ASCT+B table comprising of 14 cell types and/or anatomical structures spanning the epidermis (stratum granulosum, stratum spinosum and stratum basale) keratinocytes and dermis (glandular structures, fibroblasts, macrophages, T helper cells, T killer cells, T regs, nerve fibers and endothelial cells), as well as markers of DNA damage (p53), DNA repair (DDB2) and cell proliferation (Ki67) (summarized in **Supp. Table S2**). The rationale for choosing these biomarkers was to quantify (1) immune cell density in 3D vs 2D; (2) demonstrate 3D spatial relationships between immune cells and nearest endothelial cells; (3) measure the spatial cellular effects of aging and sun exposure on epidermis cells. Antibody information for each target protein is shown in **Supp. Table S3**.

- 6. Lines 159: The authors claim that it would have been cumbersome and time-consuming to design this study with only deep learning models. However, there are already pre-trained models that with little fine tuning provide reasonably good results (see for example StarDist3D: https://openaccess.thecvf.com/content_WACV_2020/html/Weigert_Star-convex_Polyhedra_for_3D_Object_Detection_and_Segmentation_in_Microscopy_WACV_2020_paper.html). There are also frameworks to prioritise the images that perform poorly and require better annotations (https://onlinelibrary.wiley.com/doi/10.1002/cjp2.229?utm_content=buffer3a40d&utm_medium=social&utm_source=linkedin.com&utm_campaign=buffer). This should be at least introduced in the discussion.
 - o **Response:** This is an excellent point and we have now provided more rationale for why existing pre-trained models were not used in the current study (see highlighted text below). **Results (lines 194-215):** Our method provides an integrated workflow for 2D segmentation and classification of cell types from the multiplexed images, followed by automated 3D reconstruction (see **Methods** and **Supp. Figs. 5A and B**). Cell type classification is not usually integrated into segmentation workflows, and manual thresholding/gating of biomarker signal or clustering of segmented cells is often used, which is manual and prone to errors. We developed a hybrid supervised and unsupervised segmentation/classification model where a supervised DL model was first used for 2D DAPI/nuclei segmentation, followed by unsupervised Gaussian mixture models (GMM)³⁰ for probabilistic segmentation/classification of individual cell-type (i.e., epithelial, immune) and DNA damage/repair and proliferation markers (i.e., p53, DDB2, Ki67). GMM is an excellent tool for simultaneous image normalization and detecting relative changes in biomarker intensity, allowing robust classification in each section. Combining DL and GMM provides a generalizable solution for cell segmentation and classification that works for a large datasets of whole slide images. **While there are several open source options available for cell segmentation (e.g., CellSeg³¹, Cell Profiler³² or StarDist3D³³), these would have been relatively time consuming to implement here given the large amount of data for 26 serial sections (approximately 15 GB and ~40 stitched FOV per sample**

(0.832 mm x 0.702 mm/FOV). Typically, thousands of manually annotated cells are required to develop a DL-based segmentation model, and manual annotation introduces inter- and intra-rater variability. For example, development of the CellSeg model required 29,000+ manually segmented nuclei to build a DAPI-based nuclei segmentation model³¹. Our hybrid approach is faster and more generalizable for handling larger tissue images, and it does not rely on manual tuning of image thresholding values, image normalization and morphological operations (median filtering, difference of Gaussian) and watershed algorithm parameters (such as gradient thresholding and diffusion values).

7. Lines 188-191: The authors claim that their registration approach improves accuracy in a number of scenarios. The claim should be supported by results or literature.
 - **Response:** Additional references have been included to support this point. **Discussion (lines 410-413):** *Compared to image similarity-based alignment^{14,40}, automatic block correspondences is used for initial affine alignment. Use of block correspondences improves registration speed compared to image similarity-based alignment methods^{38,40} and also accounts for local deformations that may happen during the staining and image acquisition process.*

REVIEWERS' COMMENTS:

Reviewer #1 (Remarks to the Author):

In this revised manuscript entitled "*Human Digital Twin: 3D Atlas Reconstruction of Skin and Spatial Mapping of Immune Cell Density, Vascular Distance and Effects of Sun Exposure and Aging*", Ginty and colleagues presented a 3D view of skin tissue with single-cell resolution using multiplexed imaging (MxIF). I'm pleased see the revised manuscript addressed most of my previous concerns. The authors included detailed descriptions of methods for 3D reconstruction, registration and segmentation. Furthermore, the new results on 3D spatial analysis is quite interesting. As far as I'm aware of, this analysis hasn't been done in tissue samples. Excited to see the revised version and looking forward to more coming from this group.

Though it might be not within the scope of the current study, but it would be good if the authors can comment or discuss about the theoretical limit(s) of 2D versus 3D spatial analysis. Furthermore, since the registration errors would affect the results presented in the 3D analysis, the authors should at least comments or discuss the potential caveats as well as the future directions/solutions of the issue.

Minor comments

(1). In line 221: the section size 1cm x 5um  1cm² x 5um?

(2). In Line 270: the url isn't correct and should be hubmapconsortium.github.io/vccf-visualization-2022/

Reviewer #2 (Remarks to the Author):

Thank you for addressing all my comments with detailed responses.

The addition of data comparing spatial analysis of immune and endothelial cells in 2D vs 3D strengthens the manuscript substantially. However, more discussion of this concept would be valuable. Specifically, discussing the value that 3D analysis offers the field of human tissue biology. Do you think 2D analysis underestimates the proximity of cell types in other tissues? How will 3D analysis change/improve/contribute to human tissue studies in the future?

The discussion has missed the opportunity to explain why the field of human tissue biology should invest time and effort into conducting 3D spatial analysis.